# A phenotype-based forward genetic screen identifies *Dnajb6* as a sick sinus syndrome gene

Yonghe Ding[1,2], Di Lang[3,4], Jianhua Yan[1,5], Haisong Bu[1,6], Hongsong Li[1,7], Kunli Jiao[1,5], Jingchun Yang[1], Haibo Ni[8], Stefano Morotti[8], Tai Le[9], Karl J Clark[1], Jenna Port[3], Stephen C Ekker[1], Hung Cao[9,10], Yuji Zhang[11], Jun Wang[12], Eleonora Grandi[8], Zhiqiang Li[2], Yongyong Shi[2], Yigang Li[5], Alexey V Glukhov[3], Xiaolei Xu[1]*

[1]Department of Biochemistry and Molecular Biology, Department of Cardiovascular Medicine, Mayo Clinic, Rochester, United States; [2]The Affiliated Hospital of Qingdao University & The Biomedical Sciences Institute of Qingdao University (Qingdao Branch of SJTU Bio-X Institutes), Qingdao University, Qingdao, China; [3]Department of Medicine, School of Medicine and Public Health, University of Wisconsin-Madison, Madison, United States; [4]Department of Medicine, University of California, San Francisco, San Francisco, United States; [5]Division of Cardiology, Xinhua Hospital Affiliated to Shanghai Jiaotong University School Of Medicine, Shanghai, China; [6]Department of Cardiothoracic Surgery, Xiangya Hospital, Central South University, Changsha, China; [7]Department of Cardiovascular Medicine, Jiading District Central Hospital Affiliated Shanghai University of Medicine & Health Science, Shanghai, China; [8]Department of Pharmacology, University of California, Davis, Davis, United States; [9]Department of Biomedical Engineering, University of California, Irvine, Irvine, United States; [10]Department of Electrical Engineering and Computer Science, University of California, Irvine, Irvine, United States; [11]Department of Epidemiology and Public Health, University of Maryland School of Medicine, Baltimore, United States; [12]Department of Pediatrics, McGovern Medical School, The University of Texas Health Science Center at Houston, Houston, United States

*For correspondence:
xu.xiaolei@mayo.edu

**Abstract** Previously we showed the generation of a protein trap library made with the gene-break transposon (GBT) in zebrafish (*Danio rerio*) that could be used to facilitate novel functional genome annotation towards understanding molecular underpinnings of human diseases (Ichino et al, 2020). Here, we report a significant application of this library for discovering essential genes for heart rhythm disorders such as sick sinus syndrome (SSS). SSS is a group of heart rhythm disorders caused by malfunction of the sinus node, the heart's primary pacemaker. Partially owing to its aging-associated phenotypic manifestation and low expressivity, molecular mechanisms of SSS remain difficult to decipher. From 609 GBT lines screened, we generated a collection of 35 zebrafish insertional cardiac (ZIC) mutants in which each mutant traps a gene with cardiac expression. We further employed electrocardiographic measurements to screen these 35 ZIC lines and identified three GBT mutants with SSS-like phenotypes. More detailed functional studies on one of the arrhythmogenic mutants, *GBT411*, in both zebrafish and mouse models unveiled *Dnajb6* as a novel SSS causative gene with a unique expression pattern within the subpopulation of sinus node pacemaker cells that partially overlaps with the expression of hyperpolarization activated cyclic nucleotide gated channel 4 (HCN4), supporting heterogeneity of the cardiac pacemaker cells.

## Editor's evaluation

This study presents a valuable discovery of a gene important for the function of the cardiac pacemaker. The evidence is convincing as mutation in this gene causes sick sinus syndrome (SSS) in both zebrafish and mice, and potentially in humans. This manuscript is of interest to scientists in the field of cardiology, particular cardiac electrophysiology and arrhythmia.

## Introduction

Cardiac arrhythmia affects >2% of individuals in community-dwelling adults (*Khurshid et al., 2018*). Sick sinus syndrome (SSS), also known as sinus node dysfunction or sinoatrial node (SAN) disease, is a group of heart rhythm disorders affecting cardiac impulse formation and/or propagation from the SAN, the heart's primary pacemaker. SSS manifests a spectrum of presentations such as sinus pause or arrest (SA), bradycardia, sinoatrial exit block, or tachy-brady syndrome accompanied by atrial fibrillation (AF) (*Semelka et al., 2013*; *De Ponti et al., 2018*). In addition, 20% to 60% SSS patients show abnormal response to autonomic stresses (*Dakkak and Doukky, 2020*). SSS occurs most commonly in elderly, with an estimated prevalence of 1 case per 600 adults over age 65 (*Dobrzynski et al., 2007*). Symptomatic SSS can lead to inadequate blood supply to the heart and body and contribute significantly to life-threatening problems such as heart failure and cardiac arrest. While SSS is the most common indication for pacemaker implantation worldwide (*Mond and Proclemer, 2011*), the mechanisms of SSS remain poorly understood, making it difficult to stratify SSS risk in vulnerable cohorts of patients and development of effective pharmacologic therapy for pacemaker abnormalities.

To develop mechanism-based diagnostic and therapeutic strategies for SSS, it is desirable to discover genes that are expressed in the SAN and may contribute to SSS. Unfortunately, very limited number of SSS genes and related animal models are currently available. While mutations in the cardiac sodium channel α-subunit encoding gene (*SCN5A*) (*Nof et al., 2007*; *Tan et al., 2001*) and hyperpolarization-activated cyclic nucleotide-aged channel encoding gene (*HCN4*) *Schulze-Bahr et al., 2003*; *Verkerk and Wilders, 2015* have been found to cause SSS, only a few other genes affecting the structure and/or function of the SAN were identified to increase the risk of developing SSS (*Anderson and Benson, 2010*; *Holm et al., 2011*). Classic human genetic linkage analysis-based approach has played important roles in gene discovery, but it is largely limited by the availability of suitable pedigree, especially in this age-dependent disease (*Zhu et al., 2018*). More recently, the genome-wide association studies (GWASs) have been used to identify novel genetic susceptibility factors associated with SSS (*Holm et al., 2011*; *Monfredi and Boyett, 2015*). However, owing to its statistic and associative nature, it has been difficult to confidently establish genotype-phenotype relationships for the vast amount of variants (*Lin and Musunuru, 2018*; *Tam et al., 2019*). Alternative approaches for effective identification of essential genes for SSS are thus needed.

Phenotype-based forward genetic screen in model organisms is a powerful strategy for deciphering genetic basis of a biological process. Without any *a prior* assumption, new genes can be identified that shed light on key signaling pathways. However, this approach is difficult to carry out in adult vertebrates, because of significantly increased burden of colony management efforts (*Kamp et al., 2010*; *Shen et al., 2005*). To address this bottleneck, zebrafish, a vertebrate with higher throughput than rodents, has been explored to study cardiac diseases (*Gut et al., 2017*). Despite its small body size, a zebrafish heart has conserved myocardium, endocardium, and epicardium as found in human, and adult zebrafish shows strikingly similar cardiac physiology to humans (*Bakkers, 2011*). Its heart rate is around 100 beats per minute (bpm), which is much comparable to that in human than in rodents. Adult zebrafish models for human cardiac diseases such as cardiomyopathies have been successfully generated (*Ding et al., 2020a*). Besides *N*-ethyl-*N*-nitrosourea (ENU)-based mutagenesis screens that have been conducted to identify embryonic recessive mutants, insertional mutagens such as those based on viruses and/or transposons have been developed to further increase the throughput of the screen, opening doors to screening genes affecting adult phenotypes (*Amsterdam et al., 1999*; *Wang et al., 2007*). Our team recently reported a gene-breaking transposon (GBT)-based gene-trap system in zebrafish which enables to disrupt gene function reversibly at high efficiency (>99% at the RNA level) (*Clark et al., 2011*). Approximately 1,200 GBT lines have been generated, laying a foundation for adult phenotype-based forward genetic screens (*Ichino et al., 2020*). Because the expression pattern of the affected genes in each GBT line is reported by a fluorescence reporter, we enriched

GBT lines with cardiac expression and generated a zebrafish insertional cardiac (ZIC) mutant collection (*Ding et al., 2013*). Through stressing the ZIC collection with doxorubicin, an anti-cancer drug, we demonstrated that novel genetic factors of doxorubicin-induced cardiomyopathy (DIC), such as Dnaj (Hsp40) homology, subfamily B, member 6b (*dnajb6b*), sorbin and SH3 domain-containing 2b (*sorbs2b*) and retinoid x receptor alpha a (*rxraa*), could be successfully identified (*Ding et al., 2016*; *Ding et al., 2020b*; *Ma et al., 2020*). Follow up studies on these hits confirmed their identity as important cardiomyopathy genes.

Encouraged by our success in identifying new genetic factors for DIC, we reasoned that genes for rhythm disorders could be similarly identified by directly screening adult ZIC lines using echocardiographic measurement. We had recently optimized a commercially available ECG system to define SA episodes in an adult zebrafish, and the baseline frequency of aging-associated SSS in wild-type (WT) adult zebrafish (*Yan et al., 2020*). Here, we reported a pilot screen of our ZIC collection using this ECG platform and the resultant discovery of three positive hits, followed by comprehensive expressional and functional analysis of *dnajb6b* gene that is linked to one of the hits. Together, our data prove the feasibility of a phenotype-based screening strategy in adult zebrafish for discovering new rhythm genes.

## Results

### Identification of 35 zebrafish insertional cardiac (ZIC) mutants

We recently reported the generation of more than 1200 zebrafish mutant strains using the gene-break transposon (GBT) vector (*Ichino et al., 2020*). The tagged gene in each GBT mutant is typically disrupted with 99% knockdown efficiency and its expression pattern is reported by a monomeric red fluorescent protein (mRFP) reporter *Ichino et al., 2020*. We screened 609 GBT lines based on their mRFP expression and identified 44 mutants with either the embryonic or adult heart expression *Ding et al., 2016*. Then, we outcrossed these 44 lines, aided by Southern blotting to identify offsprings with a lower copy number of insertions, *Ding et al., 2013* and identified 35 mutants with a single copy of the GBT insertion after 2–4 generations of outcross (*Table 1*; *Ding et al., 2016*). Using a combination of inverse PCR and/or 5'- and 3'-RACE PCR cloning approaches, we mapped the genetic loci of GBT inserts in these 35 mutants (*Table 1*; *Ding et al., 2013*). Most of the affected genes have human orthologs with a corresponding Online Mendelian Inheritance in Man (OMIM) number. Because each GBT line contains a single GBT insertion that traps a gene with cardiac expression, these 35 GBT lines were termed as zebrafish insertional cardiac (ZIC) mutants.

### An ECG screen of 35 ZIC lines identified three mutants with increased incidence of SA and/or AV block episodes

Because each ZIC mutant disrupts a gene with cardiac expression, we enquired whether an ECG screening can be conducted to identify genetic lesions that result in arrhythmia. Since aging is a strong risk factor for heart rhythm disorders, we initially carried our screen in 35 aged ZIC fish lines generated from incrosses to facilitate the manifestation of cardiac rhythm abnormalities (*Table 1*). Because these fish are offsprings of incrosses and have been preselected based on the mRFP tag, their genotypes consist of both heterozygous and homozygous for the affected genes. As reported recently, in WT fish aged around 2 years old, we noted baseline SA episodes in about 1 out of 20 fish (5%) fish *Yan et al., 2020*. By contrast, among the 35 ZIC mutants with mixed heterozygous and homozygous genotypes, we noted an increased incidence of SA in 3 lines, including 3 out of 13 *GBT103* mutant fish at 1.5 years old, 4 out of 10 *GBT410* mutant fish at 2 years old, and 3 out of 8 *GBT411* mutant fish at 2 years old (*Figure 1A*). In addition to SA, we also noted incidence of atrioventricular block (AVB) in 4 different *GBT103* mutant animals at 1.5 years of age. Because the increased incidence of SA and/or AVB are hallmarks of SSS, these three lines were thus identified as three candidate SSS-like mutants.

To confirm the linkage between genetic lesions and the SSS-like phenotypes, we focused on homozygous animals for further validation. Because the precise insertional positions for all the 35 ZIC lines have been mapped, all these three homozygous ZIC mutants were easily identified by genotyping PCR *Table 1*, *Figure 1B*; *Clark et al., 2011*; *Ichino et al., 2020*; *Ding et al., 2013*. In contrast to 5% WT fish whereby SA episodes can be detected, significantly increased SA incidence was noted in all three homozygous mutants at 16 months of age, with an incidence of 57.1% in the *GBT103/cyth3a*, 44.4%

**Table 1.** Collection of 35 zebrafish insertional cardiac (ZIC) mutants.

| GBT # | Gene ID | Human ortholog | Insertion position | OMIM# |
|---|---|---|---|---|
| GBT001 | casz1 | CASZ1 | 5′ UTR | 609895 |
| GBT002 | sorbs2b | SORBS2 | 1st intron | 616349 |
| GBT103 | cyth3a | CYTH3 | 1st intron | 605081 |
| GBT130 | lrp1b | LRP1 | 73rd intron | 107770 |
| GBT135 | bhlhe41 | BHLHE41 | 2nd intron | 606200 |
| GBT136 | ano5a | ANO5 | 1st intron | 608662 |
| GBT145 | epn2 | EPN2 | 1st intron | 607263 |
| GBT166 | atp1b2a | ATP1B2A | 1st intron | 182331 |
| GBT235 | lrpprc | LRPPRC | 22nd intron | 607544 |
| GBT239 | map7d1b | MAP7D1 | 1st intron | NA |
| GBT249 | b2ml | B2M | 1st intron | 109700 |
| GBT250 | ptprm | PTPRM | 1st intron | 176888 |
| GBT268 | idh2 | IDH2 | 12th intron | 147650 |
| GBT298 | zgc:194659 | NA | 1st intron | NA |
| GBT270 | zpfm2a | ZFPM2 | 2nd intron* | 603693 |
| GBT299 | dph1 | DPH1 | 1st intron | 603527 |
| GBT340 | nfatc3 | NFATC3 | 1st intron | 602698 |
| GBT345 | amot | AMOT | 1st intron | 300410 |
| GBT360 | tefm | TEFM | 1st intron | NA |
| GBT361 | abr | ABR | 3′ UTR | 600365 |
| GBT364 | mat2aa | MAT2A | 1st intron | 601468 |
| GBT386 | babam1 | BABAM1 | 2nd intron | 612766 |
| GBT402 | scaf11 | SCAF11 | 2nd intron | 603668 |
| GBT410 | vapal | VAPA | 1st intron | 605703 |
| GBT411 | dnajb6 | DNAJB6 | 6th intron* | 611332 |
| GBT412 | xpo7 | XPO7 | 1st intron | 606140 |
| GBT415 | arrdc1b | ARRDC1 | 1st intron | NA |
| GBT416 | csrnp1b | CSRNP1 | 1st intron* | 606458 |
| GBT419 | rxraa | RXRA | 1st intron* | 180245 |
| GBT422 | insrb | INSR | 6th intron | 147670 |
| GBT424 | v2rl1 | VMN2R1 | 2nd intron | NA |
| GBT425 | mrps18b | MRPS18B | 5th intron | 611982 |
| GBT503 | stat1a | STAT1 | 6th intron* | 600555 |
| GBT513 | map2k6 | MAP2K6 | 1st intron | 601254 |
| GBT589 | oxsr1b | OXSR1 | 3rd intron | 604046 |

OMIM = Online Mendelian Inheritance in Man. NA = not available.

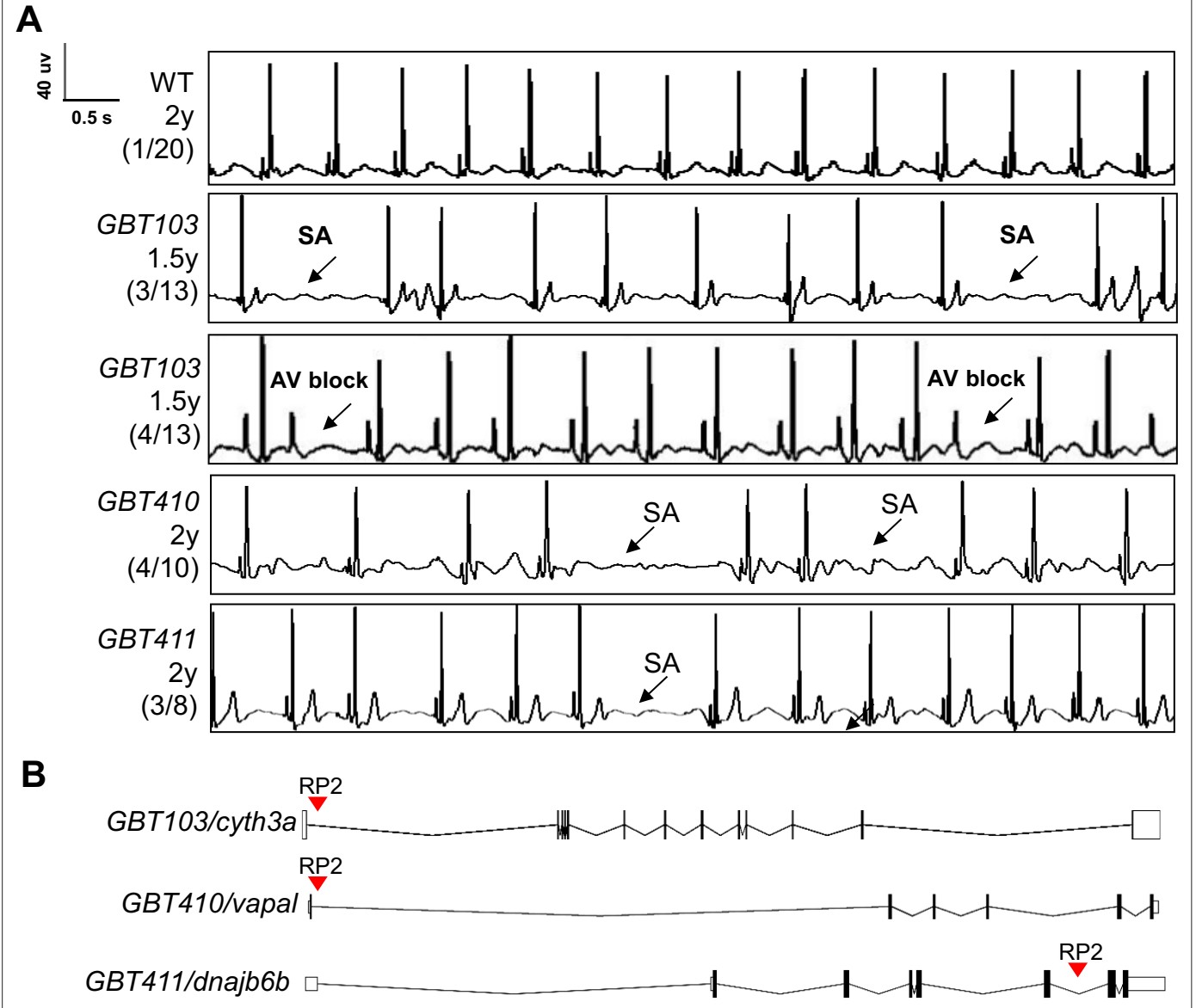

**Figure 1.** Screening of 35 ZIC lines identified three mutants with increased incidence of SA and/or AVB episodes. (**A**) Representative ECG recordings for three heterozygous/homozygous GBT mutants with increased incidence of sinus arrest (SA) and/or atrioventricular block (AVB) episodes compared to WT control. (**B**) RP2 gene-break transposon insertional positions in the three candidate SSS mutants.

The online version of this article includes the following figure supplement(s) for figure 1:

**Figure supplement 1.** The *GBT411* mutant displayed aberrant response to autonomic stimuli.

in the *GBT410/vapal*, and 40% in the *GBT411/dnajb6b* homozygous mutants, respectively (**Table 2**). There was one animal manifesting AVB in the *GBT103/cyth3a* and *GBT411/dnajb6b* homozygous mutants, respectively. In addition, we also noted a reduced heart rate, another SSS phenotypic trait in the *GBT411/dnajb6b*, but not the other two GBT homozygous mutants (**Table 2**).

To seek additional evidence supporting our screening strategy, we decided to focus on the *GBT411/ dnajb6b* mutant that is also characterized with significantly reduced heart rate phenotype. Detailed analysis of ECG indices showed increased RR interval in the *GBT411/dnajb6b* homozygous mutants (*GBT411⁻/⁻*) (**Supplementary file 2**), which is consistent with reduced heart rate. No obvious abnormality on other ECG indices was detected. Because arrhythmic mutants often manifest an aberrant response to extrinsic regulation of the heart rate, we examined responses of *GBT411⁻/⁻* to autonomic stimuli by stressing them with three compounds, including isoproterenol, a β-adrenoreceptor agonist

**Table 2.** ECG quantification to validate three GBT lines as SA mutants in homozygous fish.

| Genotype | Age | N | SA incidence (%) | AVB incidence (%) | Heart rate (bpm) |
|---|---|---|---|---|---|
| WT | 16 m | 20 | 1 (5.0) | 0 (0) | 100.1±11.1 |
| GBT103-/- | 16 m | 7 | 4 (57.1)* | 1 (14.3) | 89.1±9.1. |
| GBT410-/- | 16 m | 9 | 4 (44.4)* | 0 (0) | 99.9±17.7 |
| GBT411-/- | 16 m | 10 | 4 (40.0)* | 1 (10.0) | 90.6±7.5* |

N=7-20.

*, $p < 0.05$, data are expressed as mean ± SEM. For SA incidence comparison, Chi-square test. For heart rate comparison, unpaired student's *t*-test.

SA = sinus arrest. AVB = atrioventricular block. bpm = beats per minute.

for sympathetic nervous system; atropine, an anticholinergic inhibitor; and carbachol, a cholinergic agonist for parasympathetic nervous system. After administrating these drugs to the *GBT411$^{-/-}$* fish at 1 year old via intraperitoneal (IP) injection, we noted aberrant heart rate response to both atropine and carbachol, while its response to isoproterenol appeared to be similar to that in WT control animals (*Figure 1—figure supplement 1*).

Next, we stressed the *GBT411$^{-/-}$* fish with verapamil, an L-type $Ca^{2+}$ channel antagonists, to stress out cardiac pacemaking and unmask SSS phenotype. Indeed, SA incidence was significantly increased in the *GBT411$^{-/-}$* fish at 10 months of age (*Supplementary file 1*). Similarly, the heart rate was significantly reduced in the *GBT411$^{-/-}$* fish compared to WT controls. Together, these data provided additional evidence to support *GBT411/dnajb6b* as an arrhythmia mutant.

## Dnajb6b and its mouse ortholog exhibit unique expression patterns in the cardiac conduction system

*Dnajb6b* was previously identified as a cardiomyopathy-associated gene, disruption of which led to abnormal cardiac remodeling in zebrafish, *Ding et al., 2016* raising concerns on whether the arrhythmic phenotype in the *GBT411/dnajb6b* mutant is a primary defect in the cardiac conduction system or a consequence of cardiac remodeling in cardiomyocytes. To address this concern, we firstly defined the expression of the Dnajb6b protein in the zebrafish heart. Our previous characterization of the mRFP reporter in the *GBT411/dnajb6b* fish revealed expression of Dnajb6b protein in both the embryonic and the adult hearts *Ding et al., 2013*; *Ding et al., 2016*. To enquire its expression in the cardiac conduction system (CCS), we crossed the *GBT411/dnajb6b* line into the sqET33-mi59B transgenic line in which EGFP labels the zebrafish SAN and atrio-ventricular canal (AVC) cells (*Poon et al., 2016*). Co-localization analysis demonstrated that the mRFP positive, Dnajb6b-expressing cells partially overlap with the EGFP signal labeling both AVC and SAN cells at the base of atrium in the embryonic heart at 3 days post-fertilization (*Figure 2A-C*). In the *GBT411/dnajb6b* heterozygous (*GBT411$^{+/-}$*) adult hearts crossed with the sqET33-mi59B line, EGFP signal labeling the AVC and SAN cells were consistently detected in all animals (*Figure 2D-F*; *Poon et al., 2016*). However, in the *GBT411/dnajb6b* homozygous (*GBT411$^{-/-}$*) adult hearts crossed with the sqET33-mi59B line, EGFP signal in the AVC region appeared to be more diffused compared to that in *GBT411$^{+/-}$*, while no EGFP-positive SAN cells was detected in 3 out of 9 fish hearts examined (*Figure 2G and H*). Together, these results underscored the expression of Dnajb6b in the CCS, and disruption of *dnajb6b* in the *GBT411$^{-/-}$* mutant altered the CCS expression which might contribute to the observed SSS-like phenotypes. It should be noted that the Dnajb6b-mRFP-positive expression patterns overlap with but extend beyond the sqET33-mi59B EGFP-positive expression patterns in both embryonic and adult fish hearts (*Figure 2*).

To seek additional evidence supporting expression and function of *dnajb6b* in the CCS, we turned to the mouse model. The mouse DNAJB6 protein can be detected in all four cardiac chambers in a sectioned mouse heart tissue (*Figure 3—figure supplement 1*). Interestingly, we found a highly enriched expression of DNAJB6 in the SAN region, as defined by the expression of HCN4 channels which are responsible for the generation of hyperpolarization-activated pacemaker 'funny' current in pacemaker cells (*Figure 3A*). However, at higher magnification images, only a proportion of DNAJB6-positive cells showed colocalization with the HCN4-positive cells (arrows for colocalized cells vs. stars

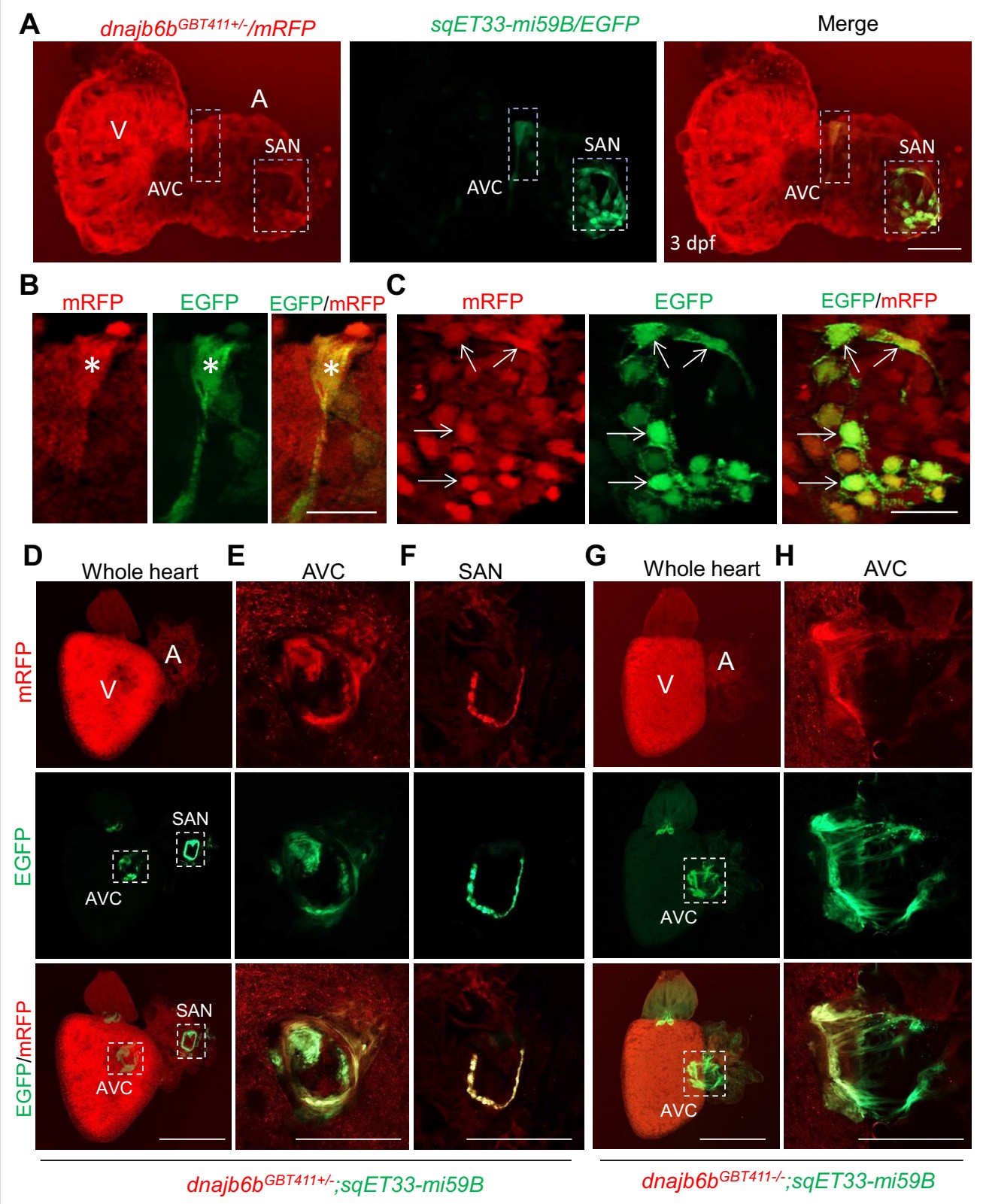

**Figure 2.** Expression and localization of Dnajb6b in zebrafish cardiac conduction system. (**A–C**) Co-localization analysis of mRFP in *GBT411/dnajb6b* heterozygous mutant with the reporter line sqET33-mi59B in which EGFP labels cardiac conduction system (CCS) in zebrafish embryos. The mRFP reporter for the *GBT411* tagged Dnajb6b protein partially overlaps with the EGFP reporter in the sqET33-mi59B transgenic line that labels atrio-ventricular canal (AVC) and sinoatrial node (SAN) in embryonic atrium at 3 dpf. Shown in (**B**) and (**C**) are higher magnification images of AVC and SAN in

*Figure 2 continued on next page*

*Figure 2 continued*

(**A**), respectively. Stars indicate EGFP + cells in the AVC, and arrows indicate EGFP + cells in the SAN. A: atrium. V: ventricle. dpf, days post-fertilization. (**D–H**) Co-localization analysis of EGFP in the *sqET33-mi59B* reporter line after crossed into the *GBT411/dnajb6b* heterozygous mutants (*dnajb6b^GBT411+/-;sqET33-mi59B*) versus *GBT411/dnajb6b* homozygous mutants (*dnajb6b^GBT411-/-;sqET33-mi59B*) in adult hearts. In the *dnajb6b^GBT411+/-;sqET33-mi59B*, EGFP is mostly expressed in the AVC within a group of confined cells, and in SAN forming a ring-like structure, which co-localizes well with mRFP. Shown in (**E**) and (**F**) are higher magnification images of AVC and SAN in (**D**), respectively. In the *dnajb6b^GBT411-/-;sqET33-mi59B*, EGFP is mostly detected in the AVC with a more diffused pattern. No ring-like structure with EGFP signal was detected in the SAN. Shown in (**H**) are higher magnification images of AVC in (**G**). Scale bars in A, 50 μm; In B, C, 20 μm; In D, G, 500 μm; In E, F, H, 200 μm.

for non-colocalized cells in *Figure 3B*). In addition, co-localization of DNAJB6 with TBX3, a transcription factor that specifies the formation of the SAN cells, was noted (*Figure 3C and D*). More interestingly, a negative correlation between DNAJB6 and TBX3 signal intensity was appreciated: cells with strong DNAJB6 expression tend to overlap with cells that show weak TBX3 signal, while cells with weak DNAJB6 expression tend to overlap with the cells with strong TBX3 signal (*Figure 3C-E*). Furthermore, the overall TBX3 signal in the SAN tissue of DNAJB6 heterozygous knock out (KO) mouse (see below) was significantly increased compared to that in WT control (*Figure 3F*). Together, these results uncovered a unique expression of DNAJB6 in the SAN region which might contribute to SSS development; however, its unique expression patterns also underscored heterogeneity of pacemaker cells within the SAN (*Boyett et al., 2000*; *Liang et al., 2021*).

## The *Dnajb6^+/-* mice manifest features of SSS when there is no sign of cardiomyopathy

To test the conservation of the cardiac arrhythmic functions of *Dnajb6b* suggested from zebrafish, we obtained a global *Dnajb6* KO mouse line. The mutant harbors a deletion of 36,843 bp nucleotides spanning from the first intron to the last intron of *Dnajb6* gene located in the Chromosome 5, which was created by the insertion of the Velocigene ZEN-Ub1 cassette and subsequent LoxP excision using Cre (*Figure 4A*). Genotyping PCR using a combination of the *Dnajb6* gene-specific and the Zen-Ubi cassette-specific primers was carried out to identify both *Dnajb6* heterozygous (*Dnajb6^+/-*) and homozygous (*Dnajb6^-/-*) KO mice (*Figure 4B*). At the protein level, both the DNAJB6 short (S) and long (L) isoforms were reduced by ~50% in *Dnajb6^+/-* mouse embryonic hearts at E12.5 stage (+/-), and near completely depleted in *Dnajb6^-/-* mutant hearts. Consistent with a previous report, *Hunter et al., 1999* *Dnajb6^-/-* KO mice were embryonic lethal, died in the uterus at about E13.5 stage, likely due to the placental defects (data not shown). The *Dnajb6* +/- mice were able to grow to adulthood without visually noticeable phenotypes until at least 1 year of age. Cardiac mechanical function remained normal, as indicated by indistinguishable cardiac echocardiography indices from those of WT siblings at the same age (*Table 3*). No abnormal myocardial structural morphology was detected in the left ventricle (LV) of *Dnajb6^+/-* mice (*Figure 4—figure supplement 1*). However, increased frequency of SA and AVB episodes, as well as bradycardia phenotype, were noted in *Dnajb6^+/-* mice at 6 months old (*Figure 4D and E*, and *Table 4*). Other ECG indices such as PR interval, QRS duration and QT interval remained comparable to WT control (*Supplementary file 3*). Similar to the *GBT411/dnajb6b* mutant in zebrafish, the *Dnajb6^+/-* mice exhibited an impaired response to autonomic stimuli including isoproterenol and carbachol (*Figure 4E*). Together, these studies suggest that *Dnajb6^+/-* mice manifest SSS phenotype without structural/functional remodeling of the heart.

## Ex vivo evidences of SAN dysfunction in the *Dnajb6^+/-* mice

To further prove SAN dysfunction in *Dnajb6^+/-* mice, we performed electrophysiological assessment of SAN pacemaker function by high-resolution fluorescent optical mapping of action potentials from isolated mouse atria at 1 year of age. We firstly analyzed the distribution of the leading pacemaker location site in *Dnajb6^+/-* mice compared to WT control. In WT mice, leading pacemakers were mostly located within the anatomically and functionally defined SAN region *Figure 5A and B*; *Gut et al., 2017*; *Glukhov et al., 2010*; *Liu et al., 2007*; *Verheijck et al., 2001*. In contrast, significant increase in the number of leading pacemakers located outside of the SAN, including the subsidiary atrial pacemakers and inter-atrial septum pacemakers, was observed in *Dnajb6^+/-* mice (p=0.039 vs. WT mice). In addition, in *Dnajb6^+/-* mice, we also found a highly irregular heart rate, accompanied by the presence of multiple competing pacemakers and a beat-to-beat migration of the leading pacemaker between

## Figure 3

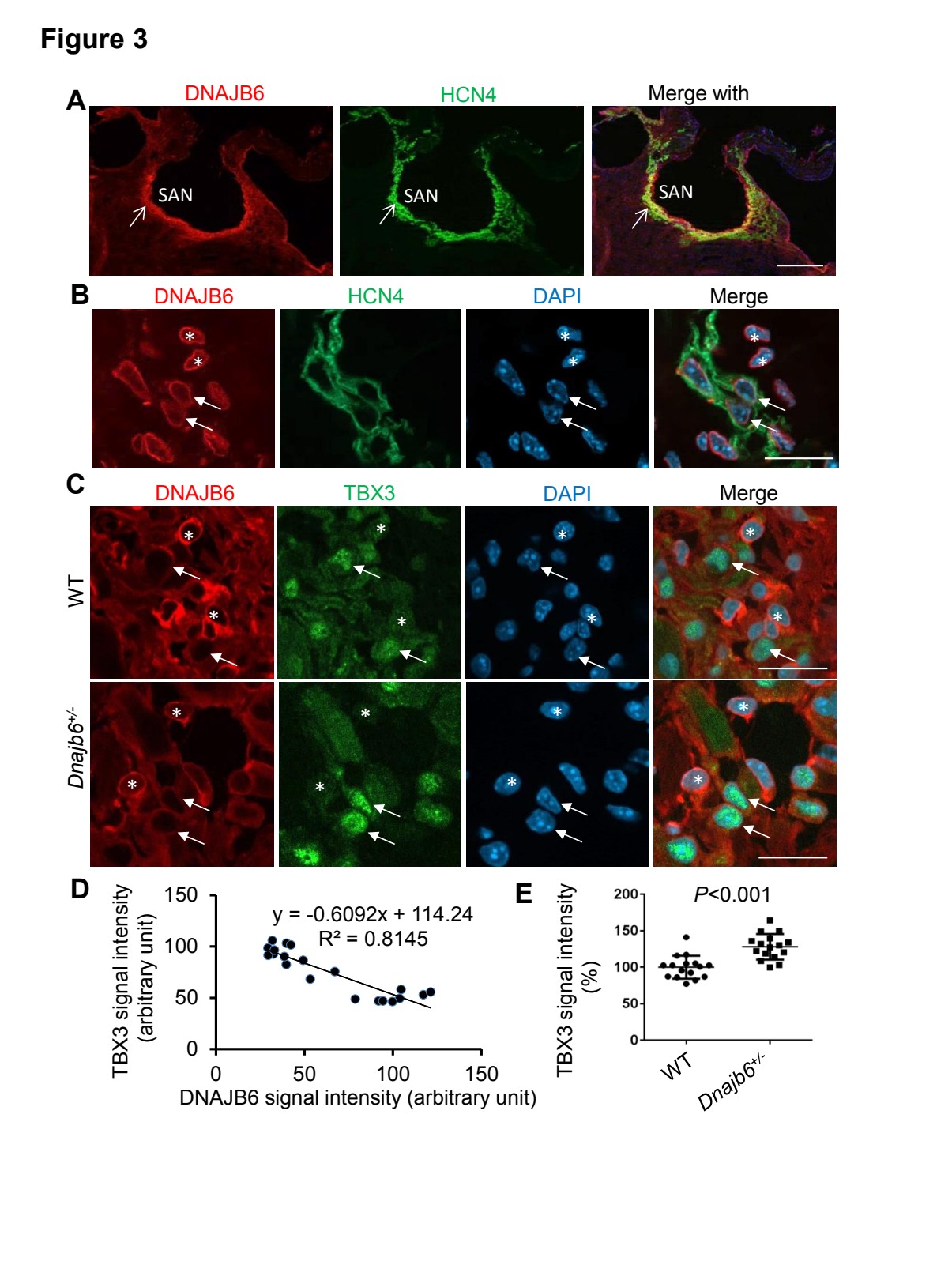

**Figure 3.** Expression and localization of DNAJB6 in the mouse SAN. (**A**) The anti-DNAJB6 antibody immunostaining signal largely overlapped with the HCN4 immunostaining signal in the mouse SAN tissues under low magnification. (**B**) Under higher magnification, expression of DNAJB6 (red) only partially overlapped with HCN4 (green) as revealed by antibody co-immunostaining. Arrows point to cells with overlapping patterns. Stars indicate cells with no-overlapping. (**C**) Shown are fluorescent images after DNAJB6 and TBX3 antibody co-immunostaining indicating expression of DNAJB6 protein

*Figure 3 continued on next page*

*Figure 3 continued*

in the WT versus *Dnajb6⁺/⁻* +/- mouse SAN. Arrows point to cells with weak DNAJB6 but strong TBX3 immunostaining signal. Stars indicate cells with strong DNAJB6 but low level of TBX3 immunostaining signal. (**D**) Quantification and correlation analysis of DNAJB6 and TBX3 immunostaining signal in WT SAN. (**E**) Quantification analysis of TBX3 signal in the WT versus *Dnajb6+/-* mouse SAN. N=20 cells. Unpaired student's *t*-test. Scale bars in A, 50 μm; In B, C, D, 20 μm.

The online version of this article includes the following figure supplement(s) for figure 3:

**Figure supplement 1.** DNAJB6 is ubiquitously expressed in 4 cardiac chambers in mouse.

various sites which included SAN, right atrial ectopic (subsidiary) pacemakers, and inter-atrial septum (***Figure 5C and D***). Similar to the results from the in vivo studies, bradycardia phenotype was consistently detected in the isolated atrial preparations as well (***Figure 5E***). Optical mapping on isolated atrial preparations further revealed different responses of heart rate during isoproterenol, atropine, and carbachol stimulations in *Dnajb6⁺/⁻* mice. Significantly increased cycle length (CL) variations were also observed at baseline and upon carbachol stimulation (***Figure 5F***).

Furthermore, in *Dnajb6⁺/⁻* mice, we found significant prolongation of the SAN recovery time corrected to beating rate (cSANRT) measured both at baseline and under autonomic stresses, including stimulation by isoproterenol, carbachol, and atropine (***Figure 5—figure supplement 1***), confirming the presence of SAN dysfunction in *Dnajb6⁺/⁻* mice. Optical mapping also showed that, unlike WT, the first spontaneous post-pacing atrial beats during SANRT measurements in *Dnajb6⁺/⁻* mice were originated from ectopic locations outside of the SAN (***Figure 5—figure supplement 1A, B***), further supporting a suppressed SAN function. Histological evaluation of fibrosis tissue content in *Dnajb6⁺/⁻* mouse atria, however, did not reveal any significant difference compared to WT mice, for both atria and SAN.

## Computational analysis of the cellular mechanisms underlying the SSS phenotype

To determine the potential cellular/ionic mechanisms underlying the observed SSS phenotype in the *Dnajb6⁺/⁻* mice , we utilized a population-based computational modeling approach. We used our previously published model of the mouse SAN myocyte to generate a population of 10,000 model variants by randomly varying selected model parameters (***Figure 6A and B***; ***Morotti et al., 2021***). In each variant, we simulated both sympathetic and parasympathetic stimulations and recorded baseline heart rate and heart rate responses to autonomic stimuli. Simulations of both our original model and population predicted an increase in heart rate with isoproterenol and heart rate slowing with carbachol. Nevertheless, the cell-cell variability in heart rate response allowed identifying two subpopulations of model variants, whereby several models (n=438) displayed a slower firing rate at baseline, an increased response to isoproterenol, and a decreased response to carbachol administration (***Figure 6C and D***), thus recapitulating the *Dnajb6⁺/⁻* mice as measured in our ex vivo functional experiments (***Figure 5E***). The WT subpopulation comprised of the remaining n=6995 models. To reveal the ionic processes that are associated with the observed electrophysiological differences in *Dnajb6⁺/⁻* vs. WT, we then compared the parameter values ( the randomly applied scaling factors) in the two model subpopulations and found significant differences in several model parameters (***Figure 6D–F***). The analysis revealed a significant decrease in the maximal conductance of the fast ($Na_v1.5$) $Na^+$ current, the L-type $Ca^{2+}$ current ($I_{Ca,L}$), the transient outward, sustained, and acetylcholine-activated $K^+$ currents, the background $Na^+$ and $Ca^{2+}$ currents, as well as the ryanodine receptor maximal release flux of the *Dnajb6⁺/⁻* vs. WT model variants. We also found a significant increase in the $Na^+/Ca^{2+}$-exchanger maximal transport rate, and conductance of the T-type $Ca^{2+}$ current and the slowly-activating delayed rectifier $K^+$ current.

## Transcriptome analysis of the *Dnajb6⁺/⁻* mutant hearts identifies altered genes encoding ion channels and proteins in the Wnt/beta-catenin pathway

To further seek molecular mechanisms underlying the SSS phenotypes observed in *Dnajb6⁺/⁻* mice, we performed whole transcriptome RNA-sequencing experiments using right atrial tissues isolated from *Dnajb6⁺/⁻* mice WT mice at 1 year of age. Transcriptomes of biological replicates for *Dnajb6⁺/⁻* mice

## Figure 4

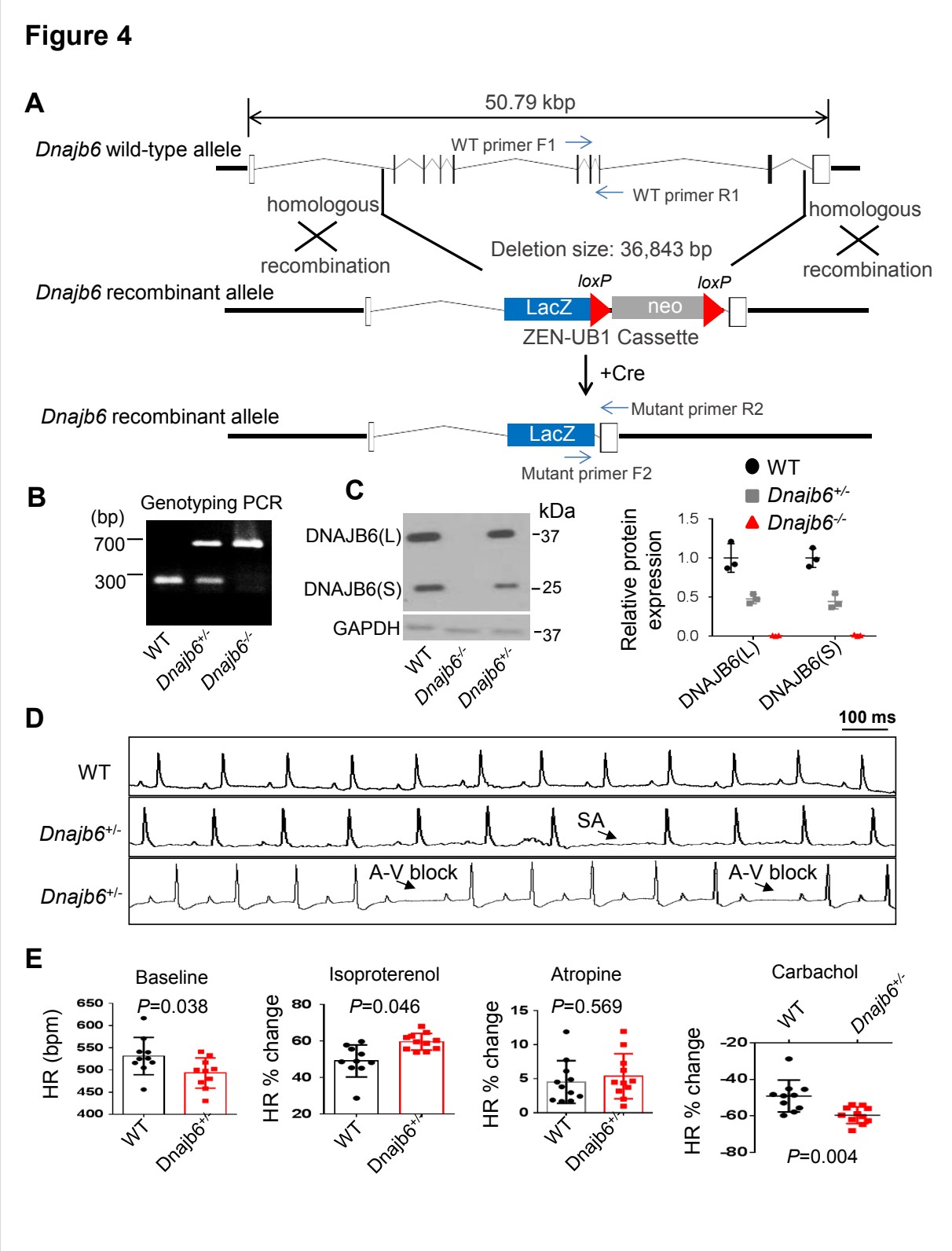

**Figure 4.** *Dnajb6*[+/-] mice exhibited increased incidence of SA and AVB and impaired response to autonomic stimuli. (**A**) Schematics of the *Dnajb6* knockout (KO) mice. The insertion of Velocigene cassette ZEN-Ub1 created a deletion of 36,843 bp nucleotides spanning from the first to the last intron of the *Dnajb6* gene at the Chromosome 5. The neomycin selection cassette was excised after crossed to a Cre expression line. (**B**) Representative DNA gel images of PCR genotyping for identifying WT (300 bp), *Dnajb6*[+/-] heterozygous (hets), and *Dnajb6*[-/-] homozygous (homo) mutant alleles . (**C**) Western

*Figure 4 continued on next page*

*Figure 4 continued*

blotting and quantification of DNAJB6 short (S) and long (L) protein expression in WT and *Dnajb6* mutants. N=3 animal per group. (**D**) Shown are representative ECG recordings results showing SA and AVB phenotypes detected in the *Dnajb6+/-* mice at 6 months. (**E**) The *Dnajb6+/-* mice manifests impaired response to different autonomic stimuli. N=10–12 mice per group, unpaired student's *t*-test. SA, sinus arrest. AVB, atrioventricular block.

The online version of this article includes the following source data and figure supplement(s) for figure 4:

**Source data 1.** Uncropped DNA gel image of PCR genotyping for identifying WT and DNAJB6 mutant mouse alleles (in PPT format).

**Source data 2.** Uncropped Western blot to show expression levels of DNAJB6 short (S) and long (L) proteins in WT and DNAJB6 mutants (in JPG format).

**Source data 3.** Uncropped Western blot to show expression levels of DNAJB6 short (S) and long (L) proteins in WT and DNAJB6 mutants (in PPT format).

**Source data 4.** Uncropped Western blot to show expression levels of DNAJB6 short (S) and long (L) proteins in WT and DNAJB6 mutants (in JPG format).

**Figure supplement 1.** No abnormal myocardium structural remodeling was detected In the *Dnajb6+/-* mice.

did form a cluster that differs from the cluster for WT control samples, as indicated by principal component analysis (PCA) (***Figure 7—figure supplement 1A***). Based on a cut-off of adjusted p-value <0.05 and≥2 folds change, 107 differentially expressed (DE) genes were identified, among which 37 genes were upregulated and 70 genes were downregulated in the *Dnajb6+/-* mice compared with WT controls (***Figure 7—figure supplement 1B, C***). Through Ingenuity pathway analysis (IPA), several diverse signaling pathways were identified to be altered in the *Dnajb6+/-* mice (***Figure 7—figure supplement 1D***). Among these 107 differentially expressed genes, we noted calcium handling related protein-encoding genes like *Slc24a2 and Cdh20*, ion channel-encoding genes including *Slc9a3r1, Kcnh7, Fxyd5, and Gjb5* (***Figure 7A***), as well as 4 Wnt pathway related genes (***Figure 7B***). We then performed quantitative RT-PCR analysis and experimentally confirmed dysregulation of these genes in the *Dnajb6+/-* mice (***Figure 7C***). The data on calcium handling and ion channel-encoding genes are in line with the SAN dysfunction phenotype observed in the *Dnajb6+/-* mice. Because Wnt signaling has been shown to direct pacemaker cell specification during SAN morphogenesis, ***Liang et al., 2020***; ***Ren et al., 2019*** the identification of 4 Wnt pathway related genes suggested that this SAN

**Table 3.** Echocardiography indices in the *Dnajb6+/-* mice compared to WT controls at 1 year.

| | WT | Dnajb6+/- | p alue |
|---|---|---|---|
| Mice number (n) | 6 | 6 | |
| HR (bpm) | 481±16 | 447±11 | 0.0017 |
| IVSd (mm) | 0.73±0.08 | 0.80±0.06 | 0.0895 |
| LVIDd (mm) | 3.92±0.33 | 3.71±0.18 | 0.2022 |
| LVPWd (mm) | 0.80±0.05 | 0.81±0.03 | 0.5204 |
| IVSs (mm) | 1.10±0.0.07 | 1.11±0.08 | 0.7878 |
| LVIDs (mm) | 2.95±0.26 | 2.77±0.15 | 0.1821 |
| LVPWs (mm) | 1.11±0.06 | 1.21±0.12 | 0.1000 |
| LVEF (%,Cube) | 57.17±5.95 | 58.17±3.92 | 0.7380 |
| LVEF (%, Teich) | 55.50±5.82 | 56.67±4.23 | 0.6996 |
| LVFS (%) | 24.67±3.61 | 25.17±2.32 | 0.7813 |
| LVd Mass (g) | 0.69±0.01 | 0.68±0.01 | 0.7650 |
| LVs Mass (g) | 0.69±0.01 | 0.69±0.01 | 1.0000 |

HR = heart rate. bpm = beats per minute. IVSd = Interventricular septum thickness at end–diastole. LVIDd = left ventricular internal dimension at end-diastole. LVPWd = left ventricular internal dimension at end-diastole. IVSs = Interventricular septum thickness at end–systole. LVIDs = Left ventricular internal dimension at end-systole. LVPWs = Left ventricular posterior wall thickness at end–diastole. LVEF = left ventricular ejection fraction. LVFS = left ventricular fractional shortening. LVd = left ventricular at end-diastole. LVs = left ventricular at end-systole. Unpaired 2-tailed student's *t*-test.

**Table 4.** ECG quantification of *Dnajb6* heterozygous mice at 6 months of age.

| Genotype | Age | N | SA incidence (%) | AVB incidence (%) | Heart rate (bpm) |
|---|---|---|---|---|---|
| WT | 6 m | 20 | 1 (5.0) | 0 | 516.3±34.3 |
| *Dnajb6+/-* | 6 m | 44 | 15 (34.1)* | 3 (6.8) | 494.8±38.3* |

N=20-44.

*, p<0.05, data are expressed as mean ± SEM. For SA incidence comparison, Chi-square test. For heart rate comparison, unpaired student's *t*-test.
SA = sinus arrest. AVB = atrioventricular block. bpm = beats per minute.

developmental pathway might play an important role in the observed SAN dysfunction phenotype in the adult *Dnajb6*[+/-] mice. Future studies are warranted to test this possibility.

## Identification of *DNAJB6* sequence variants associated with human SSS patients

To investigate the potential role of *DNAJB6* in human SSS, we queried a sequence variant dataset derived from a genome-wide association study (GWAS) of 6,469 SSS cases and 1,000,187 controls *Thorolfsdottir et al., 2021*. Out of 313 variants with minor allele frequency ≥1% in *DNAJB6*, four variants showed nominal association (p<0.05, *Supplementary file 4*). Although none of the four variants survived Bonferroni correction, it's interesting that two were located in untranslated regions. The most significant variant was observed for *rs754941044* (p=0.0193), which was predicted as a splice acceptor variant by the Ensembl variant effect predictor *McLaren et al., 2016*. Thus, it is likely this variant has a significant impact on *DNAJB6* gene function.

## Discussion

### GBT lines enable a phenotype-based screening approach for discovering new SSS genes

This work is based on recent establishment of a GBT protein trap-based insertional mutagenesis screening strategy and the generation of a collection of 1,200 zebrafish mutant strains *Ichino et al., 2020*. Here, we demonstrated the feasibility of screening these GBT lines for discovering new genetic factors for SSS, an aging-associated human disease. To overcome the challenge of colony management efforts that is associated with an adult screen, we leveraged the following unique advantages of the GBT vectors and zebrafish models. First, the knockdown efficiency for the tagged gene in each GBT homozygous mutant is consistently high, which is typically >99%, which ensued the success of an adult screen. Second, because of a fluorescence tag, heterozygous GBT fish can be easily identified under a fluorescent microscope without the need of genotyping. As a consequence, a cardiac expression-based enrichment strategy can be used to identify ZIC lines. Instead of screening 609 GBT lines, only 35 ZIC lines need to be screened, which significantly reduced the workload. We acknowledge that some genes with extremely weak cardiac expression might be missed; however, this is not a concern during the early phase of a genome-wide screen. Third, it is economically feasible to house hundreds of mutant fish lines with different genetic lesions to 1–3 years old. Finally, we optimized an ECG technology, defined the baseline SSS in WT fish, and implemented heat-stress to zebrafish at old ages, which shall increase the SSS phenotypic expressivity.

While the forward genetic screening approach has been successfully utilized to pinpoint genetic basis of cardiogenesis in embryonic fish and doxorubicin-induced cardiomyopathy (DIC) in adult zebrafish, *Amsterdam et al., 1999*; *Ding et al., 2016* this study extended this powerful genetic approach to adult zebrafish for discovering genetic factors associated with rhythm disorders. Given very little knowledge of molecular underpinnings of SSS, the development of this novel approach is significant. Human genetics approach has been difficult, partially owing to the aging associated nature - SSS-like phenotypes at its early stage are often missed, because SA episodes cannot be

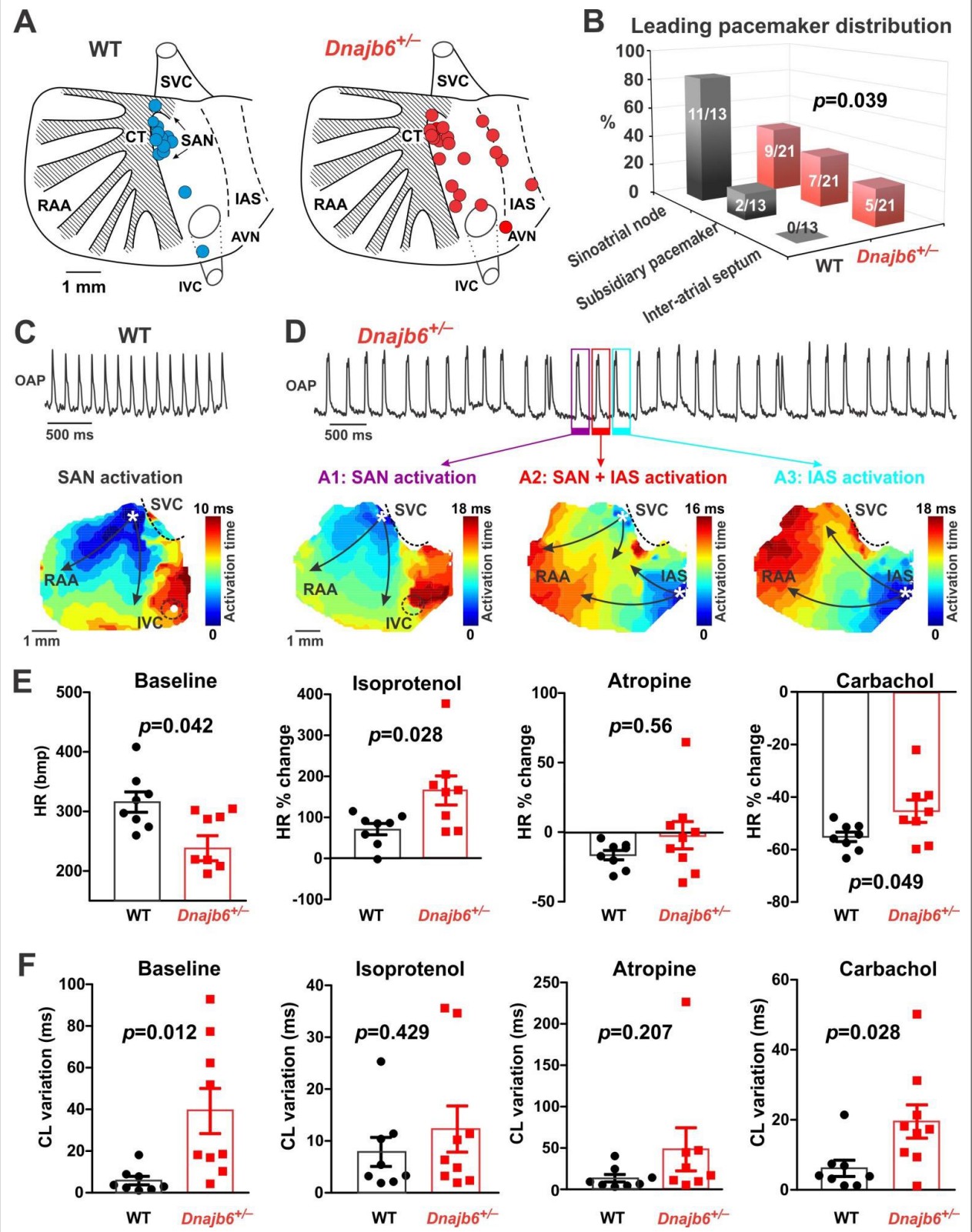

**Figure 5.** SAN dysfunction in the *Dnajb6+/-* mice. (**A**) Leading pacemakers were located and plotted from both WT (blue dots) and *Dnajb6+/-* (red dots) mice. One mouse could have multiple leading pacemaker locations due to the competing pacemakers and ectopic activities. SVC and IVC, superior and inferior vena cava; RAA, right atrial appendage; CT, crista terminalis; IAS, inter-atrial septum; AVN, atrioventricular node. Distribution of the leading pacemakers is summarized in panel. (**B**) Majority of leading pacemakers located within the SAN area in WT, whereas, in *Dnajb6+/-* mice, significant

*Figure 5 continued on next page*

*Figure 5 continued*

increase of leading pacemakers locating in subsidiary pacemaker area and IAS was observed. p-value by Fisher exact test. (**C–D**) Activation map based on the optical mapping of action potentials showed representative leading pacemaker locations in WT (SAN) and *Dnajb6*$^{+/-}$ mice (SAN and IAS areas). (**E**) Optical mapping on isolated atrial preparation showed bradycardia (baseline) and different responses of heart rate during isoproterenol, atropine, and carbachol stimulations between WT and *Dnajb6*$^{+/-}$ mice. N=7–9 mice per group. Unpaired student's *t*-test. (**F**) Increased cycle length (CL) variation was observed in *Dnajb6*$^{+/-}$ isolated atrial preparations during different autonomic stimulations. N = 7–9 mice per group, unpaired student's *t*-test.

The online version of this article includes the following figure supplement(s) for figure 5:

**Figure supplement 1.** Sinus node recovery time was prolonged in the *Dnajb6*$^{+/-}$ mice.

**Figure supplement 2.** Fibrotic tissue content in atrial and SAN myocardium of *Dnajb6*$^{+/-}$vs.WT mice.

detected if the ECG measurement only covers a short time window. It takes years in patients to develop from asymptotic to onset of SSS symptoms. Moreover, human genetic studies of SSS are typically confounded by complicated environmental factors, which are minimalized in our zebrafish forward genetic approach - each ZIC mutant is maintained in a well-controlled living environment, and the only difference among different ZIC lines is a single genetic deficiency.

## *DNAJB6* is a new SSS gene with a unique expression in SAN

The human *DNAJB6* gene encodes a molecular chaperone protein of the heat shock protein 40 (Hsp40) family. DNAJB6 has been previously linked to neurodegenerative diseases via its function in protein folding and the clearance of polyglutamine stretches (polyQ), *Gillis et al., 2013*; *Hageman et al., 2010* and to muscular dystrophy via its protein-protein interaction with Bag3 in the sarcomere *Sarparanta et al., 2012*. Our previous forward genetic screen in adult zebrafish identified *GBT411/dnajb6b* as a deleterious modifier for DIC *Ding et al., 2016*. Here, we provided several lines of evidence in both fish and mouse models, suggesting new functions of *Dnajb6* as a genetic factor for arrhythmia/SSS. First, *GBT411/dnajb6b* is one of three ZIC lines with SSS-like phenotypes that were identified from a screen of 607 GBT lines that is independent of the previous DIC screen. Second, in zebrafish, the increased incidence of SA episodes and reduced heart rate, two main features of SSS, were detected in as early as 10-month-old *GBT411/dnajb6b* homozygous fish. Similarly, bradycardia and SA episodes were noted in *Dnajb6*$^{+/-}$ KO mice at 6 months old, when the structural remodeling in both left ventricular and atrial myocardium have not occurred yet, and echocardiography indices remained indistinguishable from their age-matched siblings. Depletion of *Dnajb6* in mice manifests severer phenotypes than in zebrafish, probably because mouse has only one *DNAJB6* homologue, while zebrafish has two *DNAJB6* homologues, *dnajb6b* and *dnajb6a*. Third, consistent with loss-of-function studies, DNAJB6 expression was detected in the SAN of both zebrafish and mice. Importantly, DNAJB6 is highly enriched in the SAN region of the mouse comparing to the surrounding atrial tissue. Fourth, transcriptome analysis of *Dnajb6*$^{+/-}$ mice uncovered altered expression of genes involved in calcium handing, ion channels, and Wnt signaling pathway, which have been linked to the formation/function of the SAN during development. Thus, our data from mice strongly suggested that the observed SSS is not a consequence of *Dnajb6* cardiomyopathy, Instead, the irregular heartbeat is most likely a direct consequence of Dnajb6 depletion in pacemaker cells, subsequently contributing to the pathogenesis of cardiomyopathy that occurs later. To ultimately confirm this hypothesis and to discern functions of Dnajb6 in SAN pacemakers from working cardiomyocytes, a tissue-specific KO line for *Dnajb6* needs to be generated and studied. Prompted by our preliminary success in identifying potentially significant sequence variances for *DNAJB6* from human SSS patients, future human genetic studies are warranted to search more sequence variants and to confirm their pathogenicity, which are required to firmly establish *DNAJB6* as a new *SSS* causative gene in human.

Detailed examination of DNAJB6 expression in the SAN uncovered unique expression patterns. While the expression of DNAJB6 is detected in the SAN (*Figure 3A*), we found a partial co-expression with one of the main pacemaker protein HCN4: DNAJB6-positive cells overlap only with a portion of the HCN4-positive cells (*Figure 3B*). The unique expression pattern of DNAJB6 is also underscored by a negative correlation between the expression level of DNAJB6 and TBX3 (*Figure 3C and D*), as TBX3 is one of the main transcriptional regulators to define pacemaker cell specificity *Hoogaars et al., 2007*; *Mohan et al., 2020* While these results may sound surprising, studies on isolated SAN cells reported dramatic variability in the density of HCN4-formed 'funny' current $I_f$ *Honjo et al., 1996*; *Mangoni and Nargeot, 2001*; *Monfredi et al., 2018*; *Wilders et al., 1996*. In spontaneously beating

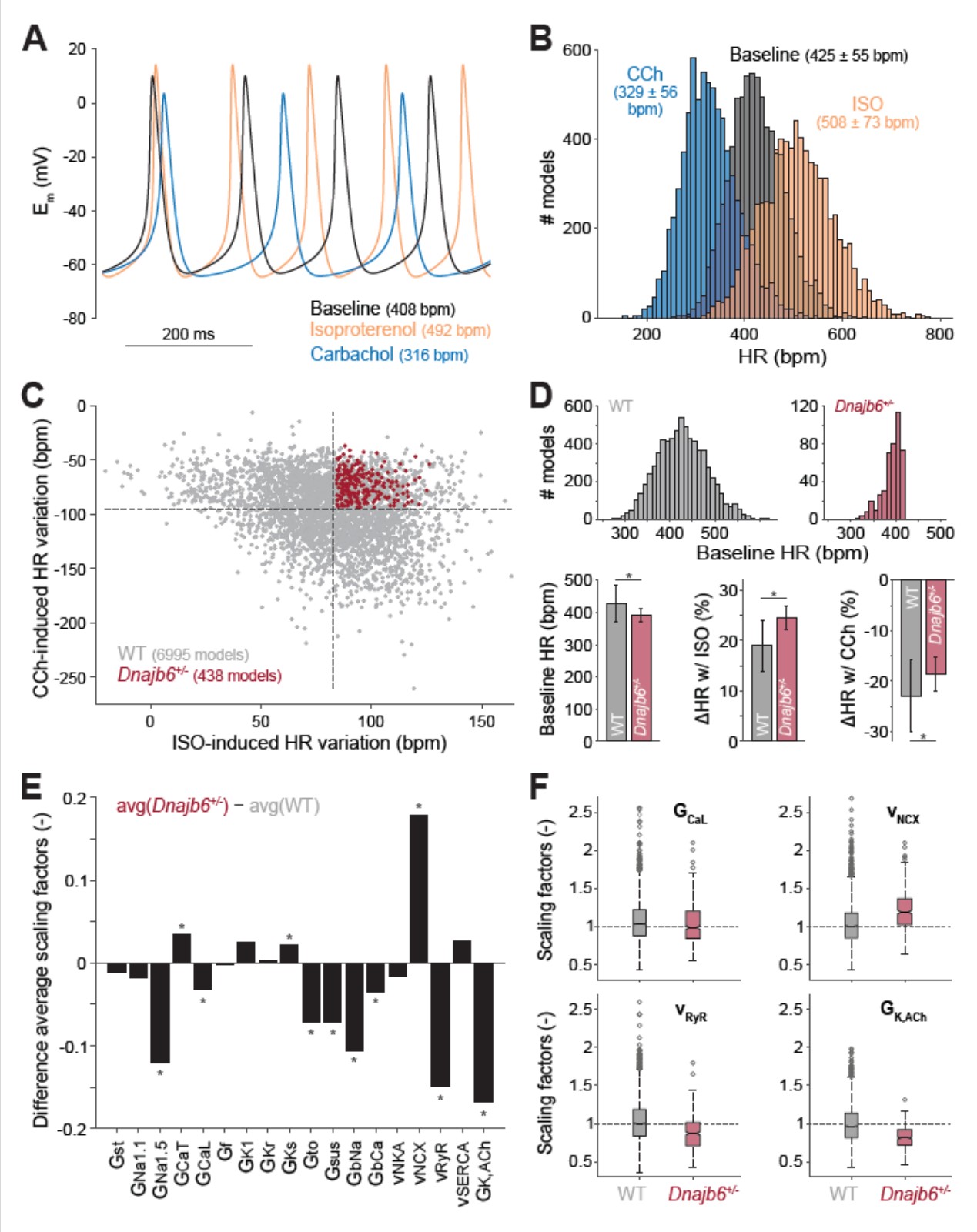

**Figure 6.** Computational analysis of the cellular mechanisms underlying the SSS phenotype observed in ex vivo mouse experiments. (**A**) Time course of membrane potential ($E_m$) predicted simulating our computational model of mouse SAN myocyte before (baseline) and after administration of isoproterenol (ISO) or carbachol (CCh). (**B**) Histogram illustrating the effects of ISO and CCh administration on firing rate (HR) distribution in our population of models. (**C**) Scatter plot quantifying HR variation in each model in the population. Red dots correspond to model variants resembling

*Figure 6 continued on next page*

*Figure 6 continued*

properties observed in ex vivo *Dnajb6*$^{+/-}$ mouse experiments ( +/- slower baseline HR, enhanced response to ISO, and reduced response to CCh), while the remaining model variants in grey mimic WT mouse functional measurements. (**D**) Histograms comparing the distribution of baseline HR in the two subgroups, and bar graphs reporting average ( ± SD) baseline HR, and relative HR variation after ISO and CCh administration in the two subgroups. (**E**) Bar graph reporting the differences between average model parameters' scaling factors in the two subgroups. Note that a positive (negative) bar corresponds to increased (decreased) average parameter value in *Dnajb6*$^{+/-}$ vs. WT groups. Asterisks in panels D and E indicate significant difference according to the 2-sided Wilcoxon rank sum test (performed with the MATALB function *ranksum*). (**F**) Statistical analysis on the values of scaling factors of selected model parameters ($G_{CaL}$, $v_{NCX}$, $v_{RyR}$, and $G_{K,ACh}$) performed with the MATLAB command *boxplot*. The central line indicates the median of each group ($q_{50}$). The central box represents the central +/- % of the data, with lower and upper boundaries corresponding, respectively, to the 25$^{th}$ and 75$^{th}$ percentiles ($q_{25}$ and $q_{75}$). The dotted vertical lines extend to 1.5 times the height of box, and individual values falling outside this range (shown here with grey circles) are considered outliers. The extremes of the lateral notches of the central box (determined as $q_{50} \pm 1.57(q_{75}-q_{25})/\sqrt{n}$, where *n* is the number of observations in each group) mark the 95% confidence interval for the medians. When the notches from two boxplots do not overlap, as in the four cases shown here, one can assume that the medians are different with a significance level of 0.05.

cardiomyocytes isolated from the rabbit SAN, Monfredi et al. showed that $I_f$ density can range from 0 to ~50 pA/pF and some the spontaneously beating SAN cells may have little to zero $I_f$ ***Monfredi et al., 2018***. The authors further observed SAN cells with lower $I_f$ current densities, demonstrating a significantly greater sensitivity to inhibition of Ca$^{2+}$ clock component of the SAN pacemaking machinery by cyclopiazonic acid, a moderate disruptor of Ca$^{2+}$ cycling, in terms of beating rate slowing. The authors also noted that a relatively large cell population (21 of 90 cells) stopped beating when the sarcoplasmic reticulum pumping rate decreased in the presence of cyclopiazonic acid, despite a relatively high $I_f$ density. Together with other studies, ***Boyett et al., 2000***; ***Kim et al., 2018*** these results may indicate a significant functional heterogeneity of pacemaker cells within the SAN in terms of their spontaneous beating rate, ion channel and calcium handling protein expression repertoire, and molecular mechanisms of their pacemaker activities. The latter was recently linked to the balance between the voltage and calcium components of the coupled-clock pacemaker system describing mechanisms of SAN automaticity ***Lakatta et al., 2010***. As summarized in details in our recent review, ***Lang and Glukhov, 2021*** it was suggested that pacemaker cells, which primary rely on the Ca$^{2+}$ clock, are more sensitive to the autonomic modulation through cAMP-mediated regulation of phosphorylation of Ca$^{2+}$ handling proteins ***Kim et al., 2018***. This is in line with our findings indicating that DNAJB6 is mainly expressed in SAN cells with low HCN4 density (***Figure 3B***) and that *Dnajb6* knock-out activates the expression of transcription factor TBX3 and affects calcium homeostasis genes (***Figure 7***) and leads to abnormal autonomic regulation of the SAN (***Figure 4E*** and ***Figure 5***).

## Potential mechanisms underlying the role of *Dnajb6* in SSS

Besides uncovering a crucial role of DNAJB6 in SAN automaticity and autonomic regulation and specification of SAN pacemaking, our studies raised several hypotheses on the underlying cellular and molecular mechanisms. Our model-based analysis provided pilot screening of potential cellular/ionic targets that could contribute to the observed SSS phenotype in *DNAJB6*$^{+/-}$ mice. Direct testing of these mechanisms would require a substantial amount of single SAN cell patch clamp and confocal microscopy experiments that can be further pursued in a follow-up study. Importantly, these new in silico experiments add another conceptual level to a phenotype-based high-throughput screening approach introduced in the current study to identify genetic factors associated with SAN dysfunction.

In addition to SSS phenotype, we observed enhanced ectopic activity in the *Dnajb6*$^{+/-}$ mice that was associated with subsidiary atrial pacemakers (***Figure 5A and B***). Based on the diffused AV canal signal and SAN signal loss in the *GBT411/dnajb6b* homozygous mutant fish hearts (***Figure 2D***), as well as a negatively correlated expression of DNAJB6 with TBX3 in the mouse SAN tissues (***Figure 3C and D***), we speculate that DNAJB6 might act as a suppressor of TBX3 transcription factor to define SAN cell specification. This potential mechanism is also supported by the observation in mice that loss-of-function of *Dnaj6b* results in conduction system defects and ectopic pacemaker activity. Since TBX3 suppresses chamber myocardial differentiation, ***Bakker et al., 2008*** upregulation of TBX3 may thus contribute to enhanced atrial ectopic activity observed in *Dnajb6*$^{+/-}$ mic. Furthermore, TBX3 has been recently identified as component of the Wnt/β-catenin-dependent transcriptional complex, ***Zimmerli et al., 2020*** which is significantly affected in *Dnajb6*$^{+/-}$ mice (***Figure 6B***). This further indicates a possible role of TBX3 in both SAN and atrial remodeling.

## A

### Six calcium homeostasis and ion channel related DE genes

| Ensemble ID | Gene | Protein | Log2 FC | P value | Gene Ontology/Function Annotation |
|---|---|---|---|---|---|
| ENSMUSG00000037996 | *Slc24a2* | Solute Carrier Family 24 Member 2 | 1.27 | 0.000623 | Cellular calcium ion homeostasis and calcium channel activity |
| ENSMUSG00000059742 | *Kcnh7* | Potassium Voltage-Gated Channel Subfamily H Member 7 | 1.21 | 8.32E-06 | Voltage-gated potassium channel activity |
| ENSMUSG00000050840 | *Cdh20* | Cadherin 20 | 1.52 | 1.37E-06 | Calcium ion binding and calcium-dependent cell-cell adhesion |
| ENSMUSG00000009687 | *Fxyd5* | FXYD-domain containing ion transport regulator | -1.01 | 0.000376 | Ion transport and sodium channel regulator activity |
| ENSMUSG00000020733 | *Slc9a3r1* | Na(+)/H(+) exchange regulatory cofactor NHE-RF | -1.19 | 0.000253 | Sodium/hydrogen exchanger regulatory cofactor |
| ENSMUSG00000042357 | *Gjb5* | Gap junction protein | -1.82 | 3.39E-05 | Gap junction channel activity |

## B

### Four Wnt pathway related DE genes

| Ensemble ID | *Gene* | Protein | Log2 FC | P value | Gene Ontology/Function Annotation |
|---|---|---|---|---|---|
| ENSMUSG00000032064 | *Dixdc1* | Dishevelled/Axin domains 1 containing protein | 1.00 | 1.77E-05 | Regulator of Wnt signaling pathway |
| ENSMUSG00000010797 | *Wnt2* | Wnt family member 2 | -1.23 | 0.000159 | Canonical Wnt signaling pathway |
| ENSMUSG00000018486 | *Wnt9b* | Wnt family member 9B | -2.27 | 3.18E-05 | Canonical Wnt signaling pathway |
| ENSMUSG00000070348 | *Ccnd1* | G1/S-specific cyclin-D1 | -1.03 | 0.000451 | A direct target gene of Wnt signaling pathway |

## C

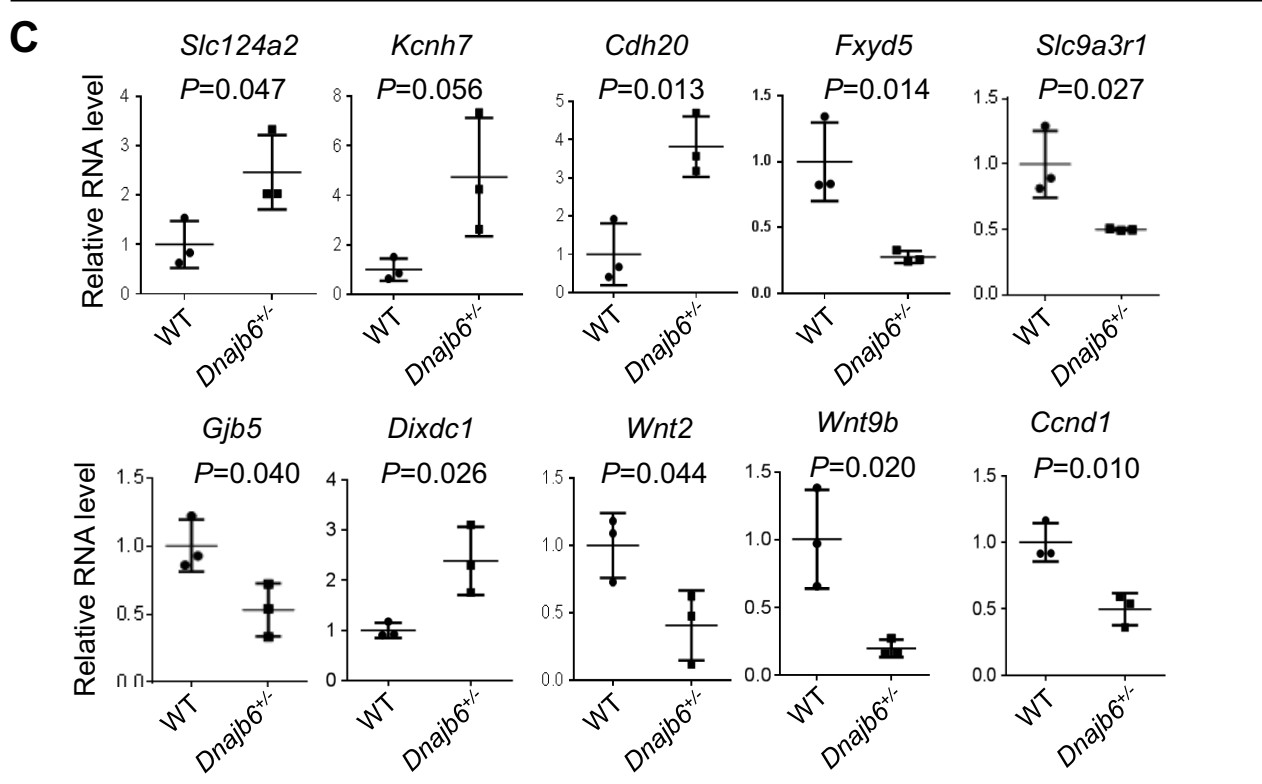

**Figure 7.** Transcriptomes are altered in the atrium of *Dnajb6+/-* mice. (**A**) Expression of six calcium homeostasis and ion channel related genes were altered in the *Dnajb6+/-* mice right atrium. (**B**) Expression of four Wnt pathway related genes were altered in the *Dnajb6+/-* mice right atrium. (**C**) Quantitative polymerase chain reaction (qPCR) validation of DE genes listed in A and B, normalized to Gapdh; RNA was extracted from an individual

*Figure 7 continued on next page*

*Figure 7 continued*

moue right atrium, which was considered a single biological replicate. Samples were collected in triplicate. N=3 mice per group, unpaired student's *t*-test.

The online version of this article includes the following source data and figure supplement(s) for figure 7:

**Source data 1.** A list of 107 differentially expressed genes identified between DNAJB6+/- knockout and WT mouse.

**Figure supplement 1.** RNA sequencing identifies transcriptome changes in the *Dnajb6⁺/⁻* mice atrium.

In the human hearts, all the observed ectopic pacemakers were located within the region of extensive distributed system of atrial pacemakers (i.e. atrial pacemaker complex), which includes but extends well beyond an anatomically defined SAN *Boineau et al., 1988*. Under physiological conditions, spontaneous activity of subsidiary pacemakers is overdrive suppressed by the SAN. However, when SAN function is diminished (i.e. during SSS), subsidiary pacemakers can produce escape beats leading to pacemaker irregularities and significant heart rate lability. Though the subsidiary pacemakers can provide a relatively regular rhythm, they are characterized by a slower resting heart, slower exertional heart rates, a prolonged post-pacing recovery time (a parameter similar to SAN recovery time but for non-SAN pacemakers), and an increased beat-to-beat heart rate variability *Morris et al., 2013*. Furthermore, the electrical activity of this subsidiary pacemakers is more akin to that of the SAN than to the surrounding atrial muscle; the subsidiary pacemaker action potential exhibits prominent diastolic depolarization and a significantly lower maximum diastolic potential, take-off potential, overshoot, rate of rise, and amplitude than typical atrial muscle *Rozanski et al., 1984*. Finally, while being bradycardic in general, subsidiary atrial pacemakers can contribute to the development of atrial tachycardia *Kistler et al., 2006*. Therefore, we hypothesize that TBX3 overexpression observed in *Dnajb6⁺/⁻* mice, could further facilitate pacemaker activity in cells within the extended atrial pacemaker complex and, maybe, promote atrial arrhythmogenesis in the setting of profound structural remodeling.

## A phenotype-based screening approach would facilitate the elucidation of molecular basis of SSS

Besides *dnajb6b*, our pilot forward genetic screen also suggested two additional candidate SSS genes like *cyth3a* and *vapal*, pending more experimental evidence to confirm their function. This forward genetic screening approach is scalable to the genome, which would generate a comprehensive list of candidate genes for SSS. Because there are at least three major cell types in the SAN region, including pacemaker cells in SAN that generate rhythm, paranodal areas and transition cells in the atrium that transmit the signal from pacemaker cells to govern coordinated contraction of the heart from atrium and then to the ventricle, *Li et al., 2017*; *Li et al., 2020* newly identified SSS genes could be categorized into different groups based on their expression pattern and phenotypic traits. We anticipate that systematic studies of these candidate genes identified from zebrafish will significantly advance our understanding of pathophysiology of SSS.

## Materials and methods
### Animals

All experiments were conducted in accordance with the Guidelines for the Care and Use of Laboratory Animals published by the US National Institutes of Health (publication No. 85–23, revised 1996). All animal procedures and protocols used in these studies (for zebrafish, #: A00005409-20; for mouse, #: A00003511-20 and M005490-R02) have been approved by the Mayo Clinic Institutional Animal Care and Use Committee (Permit number: D16-00187) and by the Animal Care and Use Committee of University of Wisconsin-Madison (Permit number: D16-00239). The zebrafish (*Danio rerio*) WIK line was maintained under a 14 hr light/10 hr dark cycle at 28.5 °C. All GBT lines were generated previously *Clark et al., 2011*; *Ichino et al., 2020*; *Ding et al., 2013*. The *Dnajb6* knockout (KO) mice, originally named *Dnajb6^tm1.1(KOMP)Vlcg*, were generated from the Jackson Laboratory (Original catalog #018623). Briefly, the insertion of Velocigene cassette ZEN-Ub1 created a deletion sized 36,843 bp nucleotides spanning from the first to the last intron of the *Dnajb6* gene at the Chromosome 5 (Genome Build37) of the C57BL/6 N mice. The mouse was subsequently bred to a ubiquitous Cre deletion mouse line for recombination of the LoxP sites that excised the neomycin selection cassette. The

following genotyping PCR primers for the *Dnajb6* mutant mice were used: mutant primer F2, 5'-AAAC TGCGCACTGTACCACC-3' and mutant primer R2, 5'-CGGTCGCTACCATTACCAGT-3' for detecting the mutant allele (predicted size of 700 bp); and WT primer F1, 5'-TACTCCAGCCCCACTCTTACTC-3' and WT primer R1, 5'- ACTGCCCATCTTCTTCAACTTC-3' for detecting the WT allele (predicted size of 300 bp).

## Enrichment and cloning of 35 ZIC mutants

Zebrafish cardiac insertional (ZIC) mutants were identified and collected based on the mRFP expression in the embryonic heart from 2 to 4 days post-fertilization (dpf) and/or in the dissected adult heart at 6 months to 1 year of age. All ZIC lines, each with a single copy of GBT insertion, were obtained after 2–4 generations of outcrosses, guided by Southern blotting using the *GFP* probe primed to the GBT vector *Ding et al., 2013*. A combination of three different methods including Inverse PCR, 5'-RACE and/or 3'-RACE were employed to clone the GBT transposon integration sites accordingly to previously published protocols *Clark et al., 2011*; *Ichino et al., 2020*; *Ding et al., 2013*. A combination of gene-specific primers flanking the GBT integration site coupled with GBT vector-specific primer were used for genotyping PCR to identify homozygous mutants for the three candidate ZIC including *GBT103/cyth3a*, *GBT410/vapal,* and *GBT411/dnajb6b* lines using genomic DNA isolated from tail fins *Clark et al., 2011*; *Ichino et al., 2020*; *Ding et al., 2013*.

## Zebrafish electrocardiogram (ECG)

Microsurgery was operated under a dissection microscope to remove the silvery epithelial layer of the hypodermis one week before fish were subjected to the ECG *Yan et al., 2020*. Fish were initially acclimated for 1 hr after transferred from the circulating fish facility to the laboratory bench, followed by anesthesia in the solution of pH 7.0 adjusted tricaine (MS-222, Sigma) at the concentration of 0.02% dissolved in E3 medium (containing 5 mM NaCl, 0.17 mM KCl, 0.33 mM CaCl$_2$, and 0.33 mM MgSO$_4$) for 6 min. Two minutes of ECG recording were then obtained with the ECG recording system, according to the instructions (ZS-200, iWorx Systems, Inc) and a recently published protocol *Yan et al., 2020*. Initial ECG screens of ZIC heterozygous mutants were performed at 32 °C using a temperature-controlled chamber set-up, made by covering the ECG recording system with a foam box. 6 to 25 fish per ZIC line were initially analyzed, depending on the fish availability. The ECG machine was held on top of a heating plate controlled by a heating machine. The subsequent ECG validation in the homozygous mutants was performed at room temperature (25 °C). To analyze the ECG recording, ECG signals were amplified and filtered at 0.5 Hz high pass and 200 Hz low pass. ECG variables, including heart rate, PR interval, QRS duration, QT interval and R-wave amplitude, and PP and RR intervals were calculated using an in-house Matlab code *Lenning et al., 2017*. A SA episode was defined in zebrafish when the PP interval is more than 1.5 s.

## Mouse ECG and echocardiography

Mouse echocardiography and ECG measurements were performed according to a previously published protocol with modifications *Ding et al., 2016*; *Ding et al., 2020b*. For ECG, mice were anesthetized with isoflurane (0.5%–1.0% v/v) via a nose cone. Mice were placed on an ECG-heater board with 4 paws on individual electrodes. The ECG-heater board maintained the body temperature at 37 °C. The ECG signal was amplified through an amplifier (Axon CNS digital 1440 A) and recorded using > Chart 5 software. For each mouse, 10 min of ECG signal were recorded. Series of ECG parameters, including heart rate, PR interval, QRS duration, QT interval and RR interval were calculated by an in-house Matlab code *Lenning et al., 2017*. For echocardiography, mice were anesthetized under light isoflurane (0.5%–1.0% v/v) administered via a nose cone. Echocardiography gel was placed on the shaved chest, and the mouse heart was imaged with a 13-MHz probe using two-dimensional echocardiography (GE Healthcare). All measurements were made by an independent operator to whom the study groups were masked.

## Administration of autonomic response drugs

For zebrafish, 0.6 µg/g isoproterenol (Millipore Sigma, Cat# 1351005), 4 µg/g atropine (Millipore Sigma, Cat# A0132), and 0.3 µg/g carbachol (Millipore Sigma, cat# C4382) were administrated via intraperitoneal injection. For in vivo mouse studies, 0.2 mg/kg isoproterenol, 1 mg/kg atropine,

and 0.3 mg/kg carbachol was injected intraperitoneally. For ex vivo mouse atrial studies, 100 nM isoproterenol, 2 µM atropine, and 300 nM carbachol was administered via superfusion for 10–20 min.

### Antibody immunostaining

Heart samples harvested from mouse SAN tissues were embedded in a tissue freezing medium, followed by sectioning at 10 µm using a cryostat (Leica CM3050 S). The slides were subjected to immunostaining using a previously described protocol *Sun et al., 2009*. The following antibodies were used: anti-HCN4 (Millipore, Cat#: AB5805; Novus biologicals, Cat#: NB100-74439) at 1:200, anti-DNAJB6 (Novus, Cat#: H00010049-M0; Santa Cruz Biotechnology Inc, Cat#: sc-104204) at 1:200, anti-TBX3 (abcam, Cat#: ab99302). All images were captured either using a Zeiss Axioplan II microscope equipped with ApoTome and AxioVision software (Carl Zeiss Microscopy) or a Zeiss LSM 780 confocal microscope. Signal intensity from DNAJB6 and TBX3 antibodies immunostaining was quantified using Zen 2.3 Pro software.

### Western blotting

For Western blotting, mice embryonic hearts at E12.5 stage were dissected after genotyping PCR using genomic DNA isolated from tail and transferred to RIPA lysis buffer supplemented with complete protease inhibitor cocktail for homogenization. About 1 µg resultant protein lysates were subject to western blotting using a standard protocol. The following primary antibodies were used: anti-Gapdh (1:4000, Santa Cruz Biotechnology Inc, Cat#: sc-25778); anti-DNAJB6 (1:6000, abcam, Cat#: 198995).

### Isolated mouse atrial preparations

The mouse atrial preparation was performed as previously described *Glukhov et al., 2015*. After the mice were anesthetized with isoflurane, a mid-sternal incision was applied. The heart was then removed and cannulated to a custom made 21-gauge cannula. The heart was then perfused and superfused with oxygenated (95% $O_2$, 5% $CO_2$), 37 °C modified Tyrode solution (in mM: 128.2 NaCl, 4.7 KCl, 1.19 $NaH_2PO_4$, 1.05 $MgCl_2$, 1.3 $CaCl_2$, 20.0 $NaHCO_3$, and 11.1 glucose; pH = 7.35 ± 0.05). Lung, thymus, and fat tissue was then removed. Perfusion was maintained under constant aortic pressure of 60–80 mmHg. After 10 min stabilization, the ventricles were dissected. The atrial were cut open as previously described *Lang and Glukhov, 2016*. The medial limb of the crista terminalis was cut to open right atrial appendage. The preparation was superfused with Tyrode solution at a constant rate of ~15 ml/min.

### Optical mapping

High spatial (100x100 pixels, 60±10 µm per pixel) and temporal (1,000–3,000 frames/sec) resolution optical mapping of electrical activity was applied on the isolated mouse atrial preparations as previously described *Lang and Glukhov, 2016*; *Lang et al., 2011*. The isolated mouse atrial preparations were coronary and surface stained with voltage-sensitive dye RH-237 (1.25 mg/ml in dimethyl sulfoxide ThermoFisher Scientific, USA). Blebbistatin (10 µM, Tocris Bioscience, USA) was then applied to reduce the motion artifact. A 150 W halogen lamp (MHAB-150W, Moritex USA Inc, CA, USA) with band pass filter (530/40 nm) was used as excitation light source. The fluorescent light emitted from the preparation was recorded by a MiCAM Ultima-L camera (SciMedia, CA, USA) after a long-pass filter (>650 nm). The acquired fluorescent signal was digitized, amplified, and visualized using custom software (SciMedia, CA, USA). After 20–30 min stabilization, activation map was collected during baseline spontaneous rhythm. To estimate the pacemaker location and a possible pacemaker shift during autonomic stimulation, the originations of action potentials were plotted with orthogonal axes crossing at the inferior vena cava. The superior to inferior direction is along the ordinate. The lateral to media direction is along the abscissa. SAN recovery time (SANRT) was measured as the time-period between the last S1S1 pacing (10 Hz) beat and the first spontaneous beat. Corrected SANRT (SANRTc) was calculated as the difference between the SANRT and the resting cycle length measure before the SANRT pacing protocol. After baseline measurement, 100 nM isoproterenol was applied. Recordings were collected after 10 min which allows the stimulation to reach steady-state effect. Complete washout was then performed which is characterized by the recovery of the heart rate back to baseline values. Additional staining and blebbistatin was applied as needed. 300 nM carbachol then was applied. 2 µM atropine was used after protocols completed during carbachol stimulation.

## RNA-seq data collection and analysis

Total RNA was extracted from the right atrium (RA) tissues of 1-year-old *Dnajb6*$^{+/-}$ heterozygous mutant hearts and WT sibling controls. Six total samples ( +/- biological replicates for each genotype) were submitted for RNA sequencing (Azenta Life Science, NJ). Genes were considered to be differentially expressed between the two groups if they exhibited a greater than twofold change and an FDR of less than 0.05 according to the DESeq approach *Love et al., 2014*. Unsupervised hierarchical clustering was performed with Pearson correlation and scaled based on the fragments per kilobase of transcript per million mapped reads (FPKM) value using the pheatmap R package (https://github.com/raivokolde/pheatmap; *Kolde et al., 2018 R Development Core Team, 2022*). The gene lists of interest were annotated by IPA (QIAGEN) (http://www.ingenuity.com/). We queried the IPA with the gene list of interest to map and generate putative biological processes/functions, networks, and pathways based on the manually curated knowledge database of molecular interactions extracted from the public literature. The enriched pathways and gene networks were generated using both direct and indirect relationships/connectivity. These pathways and networks were ranked by their enrichment score, which measures the probability that the genes were included in a network by chance.

## Quantitative reverse transcription (RT) PCR

Total RNA was extracted from ~2 mg of right atrium (RA) tissues of 1-year-old *Dnajb6*$^{+/-}$ heterozygous mutant hearts and WT sibling controls using Trizol reagent (ThermoFisher Scientific) following the manufacturer's instruction. Approximately +/- µg total RNA was used for reverse transcription (RT) and cDNA synthesis using Superscript III First-Strand Synthesis System (ThermoFisher Scientific). Real-time quantitative RT-PCR was run in 96-well optical plates (ThermoFisher Scientific) using an Applied Biosystem VAii 7 System (ThermoFisher Scientific). Gene expression levels were normalized using the expression level of glyceraldehyde 3-phosphate dehydrogenase (*gapdh*) by –ΔΔCt (cycle threshold) values. All quantitative RT-PCR primer sequences were listed in *Supplementary file 5*.

## Histology

For H&E and Masson's trichrome staining of *Dnajb6*$^{+/-}$ mice left ventricle, mouse hearts were dissected and harvested after mice were euthanized by administration of high-dose (5%) isoflurane anesthesia and after ventilation was ceased. Dissected mice hearts were immediately fixed in 4% PBS buffered formaldehyde overnight at 4 °C and sent to the Mayo Clinic Histology Core Laboratory for sample processing and H&E staining.

For transmission electron microscopy (TEM) analysis, the left ventricle apexes of dissected hearts from either zebrafish or mice were fixed immediately in Trump's solution (4% paraformaldehyde and 1% glutaraldehyde in 0.1 M phosphate buffer [pH 7.2]) at room temperature for 1 hr, followed by overnight incubation at 4 °C. Fixed samples were subsequently processed and imaged at the Mayo Clinic Electron Microscopy Core Facility using a Philips CM10 transmission electron microscope.

To quantify the amount of fibrosis in *Dnajb6*$^{+/-}$ mouse atria, the isolated atrial preparations were fixed after optical mapping experiments overnight in +/-% paraformaldehyde buffered with 0.1 M sodium phosphate, pH 7.4; and then paraffin embedded. The preparations were sectioned parallel to the epicardial surface at 3–5 µm thickness. Tissue sections were mounted on Superfrost Plus glass slides (Fisher Scientific, Pittsburgh, PA) and maintained at room temperature until use. Sections were stained for histology with Masson's trichrome (International Medical Equipment, San Marcos, CA, USA) and Picrosirius Red (International Medical Equipment, San Marcos, CA, USA). The density of fibrosis was estimated as a ratio of cardiac tissue to connective tissue measured at different transmural layers, quantified using ImageJ (National Institutes of Health) as previously described *Glukhov et al., 2015*.

## Computational modeling

To investigate the cellular mechanisms underlying the SSS phenotype, we used our model of the mouse SAN myocyte, *Morotti et al., 2021* based on the original model, *Kharche et al., 2011* and including the formulation of the acetylcholine-activated K$^+$ current developed by Arbel-Ganon et al. to simulate carbachol administration *Arbel-Ganon et al., 2020*. Functional effects of isoproterenol administration on ion channels and transporters (listed in *Supplementary file 6*) were simulated as in the parent model, *Kharche et al., 2011* wherein properties of isoproterenol-dependent modulation

of voltage-gated $Ca^{2+}$ currents and funny current $I_f$ were updated to reflect experimental observations in mice *Larson et al., 2013*; *Peters et al., 2021*. Using an established approach, *Sobie, 2009* we randomly varying selected model parameters describing maximum ion channel conductances and ion transport rates (defined in *Supplementary file 6*) to generate a population of 10,000 model variants. For each variant, the baseline value of each parameter was independently varied with a log-normal distribution ($\sigma$=0.26). We assessed the steady-state firing rate in each model in the population at baseline, and upon stimulation with either isoproterenol or carbachol. Model variants showing non-physiological behavior (e.g., lack of firing activity) at baseline or in response to autonomic stimulation were discarded from subsequent analysis. We separated the population of models in two subpopulations mimicking the WT and *Dnajb6*[+/-] mice phenotypes. Namely, we extracted the model variants that recapitulate changes observed in *Dnajb6*[+/-] vs. WT mice, including a slower firing rate at baseline, an increased response to isoproterenol, and a diminished response to carbachol administration . We analyzed the parameter value differences in these two subgroups to reveal several ionic processes that are significantly correlated with the observed electrophysiological changes. The nonparametric two-sided Wilcoxon rank sum test was used to compare the two groups and p value less than 0.05 was considered statistically significant. All the codes used to perform in silico simulations were generated in MATLAB (MathWorks, Natick, MA, USA) and are freely available for download at http://elegrandi.wixsite.com/grandilab/downloads and https://github.com/drgrandilab/Ding-et-al-2022-mouse-sinoatrial-model; *Grandi Lab, 2022* (copy archived at swh:1:rev:9ffd9fee426ef9e4b26826b4b8700a93821ba9ab).

## Statistics

No sample sizes were calculated before performing the experiments. No animals were excluded for analysis. Unpaired two-tailed student's *t*-test was used to compare two groups. One-way Analysis of Variance (ANOVA) or Kruskal-Wallis test followed by post hoc Tukey's test was used for comparing three and more groups. Chi-square test was used for rate comparison. p Values less than 0.05 was considered statistically significant. For dot plot graphs, values are displayed as mean ± standard deviation (SD). Sample size (N) represents animal number, otherwise specifically designated as biological or technical replicates. All statistical analyses were conducted with the Graphpad Prism 7 and/or R Statistical Software Version 3.6.1.

## Availability of the materials and resources

All reagents are available upon reasonable request. Zebrafish GBT mutant lines are available either from the Zebrafish International Recourse Center (ZIRC, http://zebrafish.org) or the Mayo Clinic Zebrafish Facility, respectively. Both RNAseq raw data and processed data are deposited to GEO (Access number GSE195953) associated with the token: kvmhesayzryljop. The code of our computational model of the mouse SAN myocyte is freely available for download at http://elegrandi.wixsite.com/grandilab/downloads andhttps://github.com/drgrandilab/Ding-et-al-2022-mouse-sinoatrial-model.

## Acknowledgements

We thank Beninio Gores and Kashia Stragey for managing zebrafish facility and Ronald H May for murine echocardiography. This work was supported in part by Mayo Foundation to XX, grant from Science and Technology Innovation Action Plan of Shanghai, experimental animal research project 201409005600 to YGL, NIH GM063904 to SCE, NIH R01HL141214, American Heart Association 16SDG29120011, and the Wisconsin Partnership Program 4140 to AVG, American Heart Association 17POST33370089 and American Heart Association Career Development Award 846898 to DL, NIH R00HL138160 to SM, NIH R01HL131517, NIH P01HL141084 NIH Stimulating Peripheral Activity to Relieve Conditions Grant 1OT2OD026580-01, UC Davis School of Medicine Dean's Fellow Award, and American Heart Association Scientist Development Award 15SDG24910015 to EG, and American Heart Association Postdoctoral Fellowship 20POST35120462 to HN, NIH R44OD024874 to TL and HC, NIH R01HL142704 to JW.

## Additional information

### Competing interests

Stephen C Ekker: Reviewing editor, *eLife*. The other authors declare that no competing interests exist.

### Funding

| Funder | Grant reference number | Author |
|---|---|---|
| Mayo Foundation for Medical Education and Research | | Xiaolei Xu |
| Science and Technology Innovation Action Plan of Shanghai | 201409005600 | Yigang Li |
| National Institute of Health | GM063904 | Stephen C Ekker |
| American Heart Association | 16SDG29120011 | Alexey V Glukhov |
| National Institute of Health | R00HL138160 | Stefano Morotti |
| National Institute of Health | R01HL131517 | Eleonora Grandi |
| American Heart Association | 15SDG24910015 | Eleonora Grandi |
| American Heart Association | 20POST35120462 | Haibo Ni |
| National Institute of Health | NIH R01HL141214 | Alexey V Glukhov |
| Wisconsin Partnership Program 4140 | | Alexey V Glukhov |
| American Heart Association | 17POST33370089 | Di Lang |
| American Heart Association | 846898 | Di Lang |
| National Institute of Health | 1OT2OD026580-01 | Eleonora Grandi |
| National Institute of Health | P01HL141084 | Eleonora Grandi |
| National Institutes of Health | R44OD024874 | Tai Le |
| National Institutes of Health | R44OD024874 | Hung Cao |
| National Institutes of Health | R01HL142704 | Jun Wang |

The funders had no role in study design, data collection and interpretation, or the decision to submit the work for publication.

### Author contributions

Yonghe Ding, Conceptualization, Resources, Data curation, Software, Formal analysis, Supervision, Validation, Investigation, Methodology, Writing - original draft, Writing - review and editing; Di Lang, Alexey V Glukhov, Resources, Data curation, Formal analysis, Funding acquisition, Investigation, Writing - original draft, Writing - review and editing; Jianhua Yan, Resources, Software, Formal analysis, Investigation, Methodology; Haisong Bu, Hongsong Li, Jingchun Yang, Resources, Data curation, Software, Formal analysis, Investigation, Methodology; Kunli Jiao, Resources, Data curation, Formal analysis, Investigation, Methodology; Haibo Ni, Resources, Data curation, Software; Stefano Morotti, Resources, Data curation, Software, Investigation, Methodology; Tai Le, Resources, Software, Investigation, Methodology; Karl J Clark, Stephen C Ekker, Resources, Writing - review and editing; Jenna Port, Data curation, Methodology; Hung Cao, Resources, Supervision, Writing - review and editing; Yuji Zhang, Resources, Data curation, Software, Formal analysis; Jun Wang, Conceptualization,

Investigation; Eleonora Grandi, Yongyong Shi, Supervision, Investigation; Zhiqiang Li, Data curation, Software, Formal analysis, Investigation, Methodology; Yigang Li, Resources, Supervision, Funding acquisition, Writing - review and editing; Xiaolei Xu, Conceptualization, Resources, Data curation, Software, Formal analysis, Supervision, Funding acquisition, Validation, Investigation, Visualization, Methodology, Writing - original draft, Project administration, Writing - review and editing

### Author ORCIDs
Yonghe Ding https://orcid.org/0000-0002-4531-1721
Karl J Clark https://orcid.org/0000-0002-9637-0967
Stephen C Ekker https://orcid.org/0000-0003-0726-4212
Xiaolei Xu https://orcid.org/0000-0002-4928-3422

### Ethics
All experiments were conducted in accordance with the Guidelines for the Care and Use of Laboratory Animals published by the US National Institutes of Health (publication No. 85-23, revised 1996). All animal procedures and protocols used in these studies (for zebrafish, #: A00005409-20; for mouse, #: A00003511-20 and M005490-R02) have been approved by the Mayo Clinic Institutional Animal Care and Use Committee (Permit number: D16-00187) and by the Animal Care and Use Committee of University of Wisconsin-Madison (Permit number: D16-00239).

### Decision letter and Author response
Decision letter https://doi.org/10.7554/eLife.77327.sa1
Author response https://doi.org/10.7554/eLife.77327.sa2

## Additional files

### Supplementary files
• Supplementary file 1. ECG quantification of $GBT411^{-/-}$ mutant at 10 months of age. SA, sinus arrest. epm, episode per minute. bpm, beats per minute. N=8–9. *, $P<0.05$, data are expressed as mean ± SEM. For SA incidence comparison, Chi-square test. For heart rate comparison, unpaired student $t$-test.

• Supplementary file 2. ECG quantification of $GBT411^{-/-}$ mutant at 16 months of age. N=8. *, $P<0.05$, data are expressed as mean ± SEM.

• Supplementary file 3. ECG quantification of $Dnajb6^{+/-}$ mice at 6 months of age. bpm, beats per minute. N=8 *, $P<0.05$, unpaired student's $t$-test.

• Supplementary file 4. Variants in $DNAJB6$ associated with sick sinus syndrome with $P<0.05$. [a]The effect that the variant has on each feature that it overlaps. [b]A subjective classification of the severity of the variant consequence, based on agreement with SNPEff. SNP, single nucleotide polymorphism; MAF minor allele frequency; A1, minor allele; A2, major; OR, odds ratio; SE, standard error; info, information metric (info score).

• Supplementary file 5. Quantitative RT-PCR primers used to validate differentially expressed (DE) genes identified from RNAseq.

• Supplementary file 6. Definition of model parameters and changes induced by isoproterenol (ISO) and carbachol (CCh) administration.

• Transparent reporting form

### Data availability
All data generated or analyzed during this study are included in the manuscript and supporting files. Source data files have been provided for Figure 4 and Figure 7.

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
