## [Editor Report]

This study presents a valuable discovery of a gene important for the function of the cardiac pacemaker. The evidence is convincing as mutation in this gene causes sick sinus syndrome (SSS) in both zebrafish and mice, and potentially in humans. This manuscript is of interest to scientists in the field of cardiology, particular cardiac electrophysiology and arrhythmia.

---

## [Decision Letter]

**Decision letter after peer review:**

Thank you for submitting your article "A phenotype-based forward genetic screen identifies *DNAJB6* as a sick sinus syndrome gene" for consideration by *eLife*. Your article has been reviewed by 3 peer reviewers, including Wenbiao Chen as the Reviewing Editor and Reviewer #1, and the evaluation has been overseen by Didier Stainier as the Senior Editor. The following individual involved in review of your submission has agreed to reveal their identity: Michelle Collins (Reviewer #2).

Essential revisions:

1) The manuscript needs to include new experiments that address the mechanism by which loss or reduction of DNAJB6 causes SSS.

2) The manuscript needs to have a better characterization of Dnaj6b expression in the heart and to strengthen the claim that the arrhythmia is not due to cardiomyopathy/structural remodeling in both zebrafish and mice.

3) The manuscript needs to address evidence or lack of evidence for a role of DNAJ6B in SSS in humans.

4) The revision should include the suggested recommendations on data presentation and analysis.

*Reviewer #1 (Recommendations for the authors):*

The first screen is not clearly described. While it is indicated that both heterozygous and homozygous carriers were screened, it appears that all phenotypic individuals were heterozygotes according to legend of Figure 1. Why no homozygotes were identified in the screen is not discussed. In the pilot screen, both SA and AVB were observed in the initial screen for GBT103. It is not made clear if they occur in the same individuals. In the homozygote screen, only SA incidence was assessed. It is unknown if there was no AVB or it was not assessed.

The RNAseq analysis was done in 1 year old mice. It is a stretch to link the observed changes in Wnt signaling pathway to developmental defects, which is also not supported by the histological data provided.

Since GWAS studies have been reported, can the authors find any evidence from these studies for a role of DNAJB6 in SSS?

*Reviewer #2 (Recommendations for the authors):*

Overall, this manuscript presents some exciting findings that nicely illustrate the strength of the using adult ZIC lines in a phenotypic screen for a difficult disease to model in vivo. The following recommendations are suggested to strengthen the functional studies, as some of the conclusions in the manuscript are poorly supported.

1. The claim that the zebrafish and mouse models do not have cardiomyopathy/structural remodelling is not supported by the current data. At a minimum, histology and fibrotic markers should be included. This is also important given a previous report (Ding, JCI 2016) indicated cardiomegaly in the GBT411 line. As the DNAJB6-short isoform colocalizes to the sarcomere, it would be important to show how cardiac muscle appears in GBT411/DNAJB6b hets and mutants, as well as in the DNAJB6 het mouse heart.

2. Expression patterns in zebrafish should be more clearly visualized and discussed. For example, does the mRFP from GBT411/dnajb6b colocalize with sqET33-mi59B expression in the atrioventricular canal cells at 3 dpf (only the SAN is shown), or in the adult SAN cells (only AVC is shown)? Other markers would help resolve where the DNAJB6b-mRFP cells are located in both embryonic and adult hearts (i.e. only myocardium?).

3. Electrical data from ECGs should be reported. This would support characterization of the specific conduction phenotypes (e.g. PR, QT, QRS, RR intervals). This is relevant as DNAJB6 appears quite broadly expressed in both zebrafish and mouse hearts, and so it would be interesting to see if there is only an atrial conduction issue, or if ventricular conduction parameters are also affected.

4. The presence of ectopic atrial pacemakers and the early expression of DNAJB6b could suggest that SAN patterning is affected in GBT411 mutants. Does the sqET33 line expression pattern look different between GBT411-/- and WT embryos? Markers of SAN (isl1, HCN4) could also be used to address this question during developmental stages. Along these lines, is arrhythmia detected at larval stages in GBT411 mutants?

5. Transcriptomics data seems to be centred on the ion channels and WNT pathways genes. Are these Wnt factors transcriptional regulators of the ca^2+^ channels? How is dnajb6 hypothesized to lead to transcriptional changes? Are these direct changes or a consequence of sustained SA and AVBs?

6. Have DNAJB6 variants been identified in SSS patients? Other arrhythmias?

*Reviewer #3 (Recommendations for the authors):*

1. The authors stated that DNAJB6b mutants were selected for in-depth characterization because they were most arrhythmogenic (lines 155-156). This is inconsistent with the data shown in Table 2.

2. The authors stated that DNAJB6b mutants' "response to isoproterenol remained unchanged" (lines 164-165). Based on what is presented in Supplemental Figure 1, DNAJB6b mutants responded to isoproterenol treatment in a manner like what was observed in wild type animals.

3. The cardiac expression of DNAJB6 needs further clarification.

a. The authors stated that DNAJB6 is enriched in SAN tissues (page 8). However, DNAJB6 appears to be expressed ubiquitously in the myocardium without obvious enrichment in SAN cells in fish (Figure 2).

b. Figure 2D, DNAJB6 is expressed in a broad range of cells which includes HCN4 positive cells. This is contradictory to what is stated in the abstract (lines 44-49) and discussion (lines 334-336).

c. The authors highlighted cells with high TBX3/low DNAJB6 and low TBX3/high DNAJB6 expression (Figure 2E). Could the authors explain how high/low expression is defined, how many cells have been examined and what percentage of cells showing high TBX3/low DNAJB6 and low TBX3/high DNAJB6 expression?

d. Figure 2F. islet1 staining shows high background.

4. The impacts of Dnajb6 loss on heart rhythm in mice was examined in heterozygotes because Dnajb6-/- died embryonically. There are some issues regarding these mutants that need to be clarified:

a. Could the authors clarify what stage do the homozygotes die and what stage are the hearts taken from mutants for Western blot? (Lines 213-216, Figure 3C).

b. In mice, sinus arrest and AV block were noted in Dnajb6 +/-. Could the authors provide quantification? How many animals were examined? What percentage of animals develop sinus arrest and what percentage with AV block?

c. Optical mapping revealed ectopic pacemaker activity in Dnajb6 +/- hearts. Could authors provide quantitative data?

5. What is the cellular basis of ectopic pacemaker activity in dnajb6 mutants? Are pacemaker cells damaged in Dnajb6 mutants?

6. Transcriptomic profiling.

a. Figure 6A. Missing log2FC value and p value for Gjb5.

b. Heatmap shown in sup. Figure 3B is quite unusual. Is it scaled?

---

## [Author Response]

Essential revisions:(1) The manuscript needs to include new experiments that address the mechanism by which loss or reduction of DNAJB6 causes SSS.

We carried out the following experiments to address the cellular and molecular mechanisms underlying the observed SSS phenotypes. Based on these data, we added a new section to the Discussion section (Lines 425-466).

(1) In mice, we did more antibody immunostaining and confirmed a negative correlation in terms of expression intensity between the DNAJB6 and TBX3 proteins. We further detected a significantly increased TBX3 immunostaining signal in the SAN tissues of *DNAJB6* heterozygous mice compared to WT controls (new Figure 3D-F).

(2) In zebrafish, we compared expression patterns of the sqET33-mi59B conduction system reporter line between the *GBT411/dnajb6b* heterozygous and homozygous mutants. We found the atrio-ventricular canal (AVC) signal became diffused in *GBT411/dnajb6b* homozygous adult hearts. In addition, the ring-like structure usually seen in the SAN region of WT controls and in the *GBT411/dnajb6* heterozygous was largely lost in 3 out of 9 *GBT411/dnajb6b* homozygous adult hearts examined (new Figure 2).

Together with the ectopic pacemaker activity detected in the *Dnajb6* heterozygous mice (new Figure 5A and 5B), we speculate that DNAJB6 might act as a suppressor of TBX3 transcription factor in defining cell fate specification into SAN pacemaker myocytes. Since TBX3 was reported to suppress chamber myocardial differentiation (Mommersteeg et al., Circ Res. 2007;100(3):354-62), upregulation of TBX3 may thus contribute to enhanced atrial ectopic activity in *Dnajb6* heterozygous mice.

Furthermore, TBX3 has been recently identified as a component of the Wnt/β-catenin-dependent transcriptional complex (Zimmerli et al., *eLife*. 2020;9:e58123), which is significantly affected in Dnajb6 heterozygous mice (see new Figure 7B-C). This further supports a possible role of TBX3 in both SAN and atrial remodeling.

(3) Finally, in collaboration with Drs. Grandi, Morotti, and Ni from University of California Davis, we utilized a population-based computational modeling approach to determine the cellular/ionic mechanisms that could underlie the ex vivo observed SSS phenotype in the *Dnajb6* heterozygous mice (new Figure 6). We used our previously published model of the mouse SAN myocyte (Morotti et al. Int J Mol Sci. 2021; 22(11):5645) and enhanced it with addition of both sympathetic and parasympathetic stimulations to model the effects of isoproterenol- and carbachol-induced changes in pacemaker activity (i.e., firing rate), respectively. We generated a population of 10,000 mouse SAN myocyte models by random modification of selected model parameters describing maximum ion channel conductances and ion transport rates from the baseline model and assessed isoproterenol- and carbachol-induced effects on each model variant. We then separated this population of models in two subpopulations representing the WT and *Dnajb6^+/-^* mice phenotypes: namely, we extracted the model variants that recapitulate changes observed in *Dnajb6^+/-^* vs. WT mice, including a reduced firing rate at baseline, an increased response to isoproterenol, and a decreased response to carbachol administration (new Figure 6). This filtering process resulted in n=438 models that correspond to the *Dnajb6^+/-^* mice phenotype and n=6,995 models that correspond to the WT phenotype. We analyzed the parameter value differences in these two subgroups to revealed several crucial parameters that are significantly correlated with the observed electrophysiological changes. The analysis revealed a significant decrease in the maximal conductances of the fast (Na_v_1.5) sodium current, the L-type ca^2+^ current (*I*_Ca,L_), the transient outward, sustained, and acetylcholine-activated K^+^ currents, the background Na^+^ and ca^2+^ currents, as well as the ryanodine receptor maximal release flux of the *Dnajb6^+/-^* vs. WT model variants. We also found a significant increase in the Na^+^/ca^2+^ exchanger (NCX) maximal transport rate, and conductances of the T-type ca^2+^ current and the slowly-activating delayed rectifier K^+^ current. These new studies provide some novel mechanistic insights into the observed SSS phenotype in *Dnajb6^+/-^* mice. Importantly, these new *in silico* experiments add another conceptual level to the phenotype-based screening approach introduced in the current study to identify new genetic factors associated with SAN dysfunction. Direct testing of these mechanisms would require a substantial amount of single SAN cell patch clamp and confocal microscopy experiments which are out of scope of the current manuscript and will be pursued in a follow-up study.

(2) The manuscript needs to have a better characterization of Dnaj6b expression in the heart and to strengthen the claim that the arrhythmia is not due to cardiomyopathy/structural remodeling in both zebrafish and mice.

(1) In zebrafish, we obtained higher quality images of *GBT411/sqET33-mi59B*, demonstrating partially overlapping expression of the EGFP reporter in the sqET33-mi59B line with the dnajb6b-mRFP fusion protein in the *GBT411* mutants during both embryonic and adult stages (new Figure 2).

(2) In mice, we obtained higher resolution images of antibody immunofluorescence staining in the *Dnajb6^+/-^* mice and demonstrated that DNAJB6 is partially co-localized with the HCN4 protein and showed a complementary expression with TBX3 in the SAN (new Figure 3).

(3) In mice, we also carried out more histology analysis experiments to examine left ventricular (LV) structure in *Dnajb6^+/-^* mice at 1 year of age, using H&E staining, Masson’s Trichrome staining, and transmission electron microscopy (TEM) analysis. We now show clearly that there are no significant myocardium structural changes in the LV tissues of *Dnajb6^+/-^* mice, when the SSS phenotype was already noticeable (new Figure 4—figure supplement 1). In addition, we did more detailed fibrotic contents analysis in the atrial and SAN tissues of *Dnajb6^+/-^* mice at 1 year of age and detected no significant changes as well (new Figure 5—figure supplement 2). Together, these new data further confirm the absence of structural remodeling in *Dnajb6^+/-^* mice and highlight a direct effect of *Dnajb6* on SAN dysfunction.

(4) Finally, in zebrafish, we examined myocardium ultrastructure in the *GBT411/dnajb6b* heterozygous fish at ~2 years of age by using TEM analysis. We detected severe sarcomere structural changes in 1 out of 3 fish hearts examined (see Author response image 1). Because both arrhythmia and myocardium structural abnormality were detected in fish at this age, this experiment cannot be used to support the statement that arrhythmia is not due to structural remodeling. An earlier time point will be needed in the future.

**Author response image 1. sa2fig1:** Myocardium structural defects were detected in the *GBT411* heterozygous mutant at 2 years of age. Shown are transmission electron microscopy micros (TEM) images of WT and *GBT411* heterozygous mutant heart ventricles at 2 years of age. Arrows point to round-shape and swollen mitochondrial. Stars indicate degenerated sarcomeric Z-discs. Scale bars, 5µm.

Together, we have obtained more experimental evidence to strengthen the claim that arrhythmia is not due to cardiomyopathy/structural remodeling in the *Dnajb6^+/-^* mice. However, the evidence from fish remains weak. Therefore, we removed the claim that “when structural remodeling/cardiac dysfunction have not yet occurred” in fish and modified our statement in mice accordingly (Lines 375-378, 386-389).

(3) The manuscript needs to address evidence or lack of evidence for a role of DNAJ6B in SSS in humans.

(1) From a recent GWAS study based on 6,469 SSS cases and 1,000,187 controls, a dataset for genetic variants for SSS have been reported (Thorolfsdotir et al., Eur Heart J, 2021;42(20):1959-1971). We searched DNAJB6 variants from this database and identified four variants with P<0.05, among which two are located in untranslated regions. One variant, rs754941044 (P=0.0193), is predicted as a splice acceptor variant by the Ensembl variant effect predictor, which is likely to have a significant impact on DNAJB6 gene function (new Supplementary File 4). We added this data to the main text (Lines 320-328).

(2) We performed targeted genetic screening of the translated sequence of DNAJB6 in DNA of 162 human SSS patients enrolled in Xinhua hospital, Shanghai Jiaotong University School of Medicine, Shanghai, China, for possible disease association. We identified one variant from one patient, as shown in the Author response table 1.

**Author response table 1. sa2table1:** Identification of the *DNAJB6* variant from patients with sick sinus syndrome (n=162).

Nucleotide change (protein change)	Genotyoe frequency	ExAC MAF frequency (%)	Clinical phenotype	SIFT prediction (score)	PolyPhen Prediction (score)
c.145G>T (p.V49L)	0/8600 (EVS)1/95768 (ExAC)1/140242 (GnomAD)1/162 (SSS)	0	Sick Sinus syndrome	Tolerate (o.142)	Damaging (0.972)

EVS, exome variant server; ExAC, exome aggregationconsortiu; GnomAD, gemone aggregation database; MAF, minor allele frequency; SIFT, sorting intolerant; SSS, sick sinus syndrome

Because our current evidence is still relatively weak – more pathogenic variants would need to be identified and functional validation of their pathogenicity need to be performed, we decided to not include these data in the present manuscript.

Taken together, in the Discussion section, we state that “Prompted by our preliminary success in identifying potentially significant sequence variances for DNAJB6 from human SSS patients, future human genetic studies are warranted to search more sequence variants and to confirm their pathogenicity, which are required to firmly establish DNAJB6 as a SSS causative gene in human.” (Lines 391-395)

(4) The revision should include the suggested recommendations on data presentation and analysis.

Thanks to reviewer’s suggestions on our data presentation. We carefully revised the manuscript accordingly. Below please see our point-to-point response to reviewers’ comments, and suggested recommendations on data presentation and analysis as well.

Reviewer #1 (Recommendations for the authors):The first screen is not clearly described. While it is indicated that both heterozygous and homozygous carriers were screened, it appears that all phenotypic individuals were heterozygotes according to legend of Figure 1. Why no homozygotes were identified in the screen is not discussed. In the pilot screen, both SA and AVB were observed in the initial screen for GBT103. It is not made clear if they occur in the same individuals. In the homozygote screen, only SA incidence was assessed. It is unknown if there was no AVB or it was not assessed.

(1) We initially performed the pilot screen in aged, mRFP positive GBT mutant fish generated from incrosses, thus containing both heterozygous and homozygous carriers. For our initial screen, we did not breed all GBT lines to homozygosity purposely because of the significantly increased colony management efforts. After candidate GBT lines with SSS phenotypes have been identified, we then selectively bred these GBT lines to homozygous animals. We now clarified this point in the text (Lines 133-134, 138-142).

(2) We now confirm that SA and AVB were detected in different animals of the *GBT103* mutants mixed with heterozygous and homozygous in the initial pilot screen. In the homozygote confirmation experiments, there was only 1 out of 7 *GBT103* homozygous fish (15%), and 1 out of 10 *GBT411* homozygous fish (10%) manifesting AVB. We added these AVB data and revised the Table 2 accordingly and clarified these details in the main text (Lines 142-143, 153-154).

The RNAseq analysis was done in 1 year old mice. It is a stretch to link the observed changes in Wnt signaling pathway to developmental defects, which is also not supported by the histological data provided.

We agree that it might be a stretch to conclusively link Wnt signaling pathway with *Danjb6* mutant phenotypes. However, it is a very intriguing data that RNAseq data uncovered a change in Wnt signaling that have been linked to SAN development. We toned down our conclusions and pointed out this stretch, so that readers can properly interpret these data (Lines 315-319).

Nevertheless, we detected a negative correlation in terms of expression intensity between the DNAJB6 and TBX3 proteins and further found a significantly increased TBX3 signal in the SAN tissues of *Dnajb6* heterozygous mice compared to WT controls (new Figure 3). Since TBX3 suppresses chamber myocardial differentiation (Circ Res. 2007;100(3):354-62), upregulation of TBX3 may thus contribute to enhanced atrial ectopic activity observed in *Dnajb6* heterozygous mice. These findings provide some mechanistic insights into the observed arrhythmic phenotype and highlight specific signaling pathways that need to be further addressed in future studies.

Since GWAS studies have been reported, can the authors find any evidence from these studies for a role of DNAJB6 in SSS?

(1) From a recent GWAS study based on 6,469 SSS cases and 1,000,187 controls, a dataset for genetic variants for SSS have been reported (Thorolfsdotir et al., Eur Heart J, 2021;42(20):1959-1971). We searched *Dnajb6* variants from this database and identified four variants with *P*<0.05, among which two are located in untranslated regions. One variant, *rs754941044* (*P*=0.0193), is predicted as a splice acceptor variant by the Ensembl variant effect predictor, which is likely to have a significant impact on *Dnajb6* gene function (new Supplementary File 4). We added this data to the main text (Lines 320-328).

(2) We performed targeted genetic screening of the translated sequence of *Dnajb6* in DNA of 162 human SSS patients enrolled in Xinhua hospital, Shanghai Jiaotong University School of Medicine, Shanghai, China, for possible disease association. We identified one variant from one patient, as shown in the Response-only Table 1. Because our current evidence is still relatively weak – more pathogenic variants would need to be identified and functional validation of their pathogenicity need to be performed, we decided to not include these data in the present manuscript.

Taken together, in the Discussion section, we stated that “Prompted by our preliminary success in identifying potentially significant sequence variances for *Dnajb6* from human SSS patients, future human genetic studies are warranted to search more sequence variants and to confirm their pathogenicity, which are required to firmly establish *Dnajb6* as a new human *SSS* causative gene.” (Lines 391-395)

Reviewer #2 (Recommendations for the authors):Overall, this manuscript presents some exciting findings that nicely illustrate the strength of the using adult ZIC lines in a phenotypic screen for a difficult disease to model in vivo. The following recommendations are suggested to strengthen the functional studies, as some of the conclusions in the manuscript are poorly supported.1. The claim that the zebrafish and mouse models do not have cardiomyopathy/structural remodelling is not supported by the current data. At a minimum, histology and fibrotic markers should be included. This is also important given a previous report (Ding, JCI 2016) indicated cardiomegaly in the GBT411 line. As the *Dnajb6*-short isoform colocalizes to the sarcomere, it would be important to show how cardiac muscle appears in GBT411/dnajb6b hets and mutants, as well as in the Dnajb6 het mouse heart.

We carried out more experiments to examine left ventricular (LV) structure in *Dnajb6* heterozygous mice at 1 year of age, using H&E staining, Masson’s Trichrome staining, and transmission electron microscopy (TEM) analysis. We now show clearly that there are no significant myocardium structural changes in the LV as well as atrial and SAN tissues of *Dnajb6* heterozygous mice (new new Figure 4—figure supplement 1, Figure 5—figure supplement 2), when the SSS phenotype was already noticeable. However, in the *GBT411/dnajb6b* heterozygous mutant at ~2 years of age, we detected severe sarcomere structural abnormality in 1 out of 3 fish hearts examined (see Response-only Figure 1). In addition, in a previous publication (Ding et al., Circ Res, 2013:112(40:606-17)), we reported evident cardiac remodeling phenotypes in the *GBT411/dnajb6b* homozygous fish at 12 months of age.

Together, we have obtained more experimental evidence to strengthen the claim that arrhythmia is not due to cardiomyopathy/structural remodeling in the *Dnajb6^+/-^* mice. However, the evidence from fish remains weak. Therefore, we removed the claim that “when structural remodeling/cardiac dysfunction have not yet occurred” in fish and modified our statement in mice accordingly (Lines 375-378, 386-389).

2. Expression patterns in zebrafish should be more clearly visualized and discussed. For example, does the mRFP from GBT411/dnajb6b colocalize with sqET33-mi59B expression in the atrioventricular canal cells at 3 dpf (only the SAN is shown), or in the adult SAN cells (only AVC is shown)? Other markers would help resolve where the Dnajb6b-mRFP cells are located in both embryonic and adult hearts (i.e. only myocardium?).

Thanks for these suggestions. We carried out more experiments to visualize and compare the mRFP patterns in the *GBT411/Dnajb6b* mutants to the EGFP expression in the conduction system reporter line of sqET33-mi59B. Our new data clearly showed that *GBT411/mRFP* colocalized with the sqET33-mi59B/EGFP in both atrioventricular canal (AVC) cells and SAN in the atrium, in both embryonic (3 dpf) and adult hearts (new Figure 2).

3. Electrical data from ECGs should be reported. This would support characterization of the specific conduction phenotypes (e.g. PR, QT, QRS, RR intervals). This is relevant as Dnajb6 appears quite broadly expressed in both zebrafish and mouse hearts, and so it would be interesting to see if there is only an atrial conduction issue, or if ventricular conduction parameters are also affected.

For both mice and fish, we added quantification data for other ECG parameters including PR interval, QT interval, QRS duration and RR interval as suggested (new Supplementary Files 2 and 3). Our quantification detected no statistically significant changes, except for increased RR intervals in mutants, which is consistent with the observed heart rate reduction.

4. The presence of ectopic atrial pacemakers and the early expression of Dnajb6b could suggest that SAN patterning is affected in GBT411 mutants. Does the sqET33 line expression pattern look different between GBT411-/- and WT embryos? Markers of SAN (isl1, HCN4) could also be used to address this question during developmental stages. Along these lines, is arrhythmia detected at larval stages in GBT411 mutants?

We didn’t detect arrhythmia phenotypes at larval stages in the *GBT411* mutants. The late-onset and milder phenotypes in zebrafish than mouse is probably owning to the existence of two *Dnajb6* homologues in zebrafish, but a single *Dnajb6* homologue in mouse.

Thanks to your question, we compared more carefully the expression patterns of sqET33 line expression patterns between *GBT411/dnajb6b* heterozygous and homozygous mutants and WT control in both embryonic and adult hearts. We didn’t see obvious differences in embryonic hearts. However, in adult hearts, we found that the AVC signal became diffused and SAN signal was largely lost in 3 out of 9 *GBT411/dnajb6b* homozygous mutant hearts examined comparing to *GBT411/dnajb6b* heterozygous mutant (new Figure 2).

5. Transcriptomics data seems to be centred on the ion channels and WNT pathways genes. Are these Wnt factors transcriptional regulators of the ca^2+^ channels? How is DNAJB6 hypothesized to lead to transcriptional changes? Are these direct changes or a consequence of sustained SA and AVBs?

Our RNAseq data suggested that *Dnajb6* might transcriptionally regulate expression of genes in Wnt pathway and ca^2+^ channel, providing clues for future mechanistic studies. At this stage, we do not have evidence supporting that Wnt factors are transcriptional regulators of the ca^2+^ channels. Because the RNAseq was done using 1 year mice when SSS is obvious, it is less likely that these transcriptional changes are direct consequences of *Dnajb6* gene knockout. Longitudinal studies are needed in the future to answer these mechanistic questions.

6. Have DNAJB6 variants been identified in SSS patients? Other arrhythmias?

(1) From a recent GWAS study based on 6,469 SSS cases and 1,000,187 controls, a dataset for genetic variants for SSS have been reported (Thorolfsdotir et al., Eur Heart J, 2021;42(20):1959-1971). We searched *DNAJB6* variants from this database and identified four variants with *P*<0.05, among which two are located in untranslated regions. One variant, *rs754941044* (*P*=0.0193), is predicted as a splice acceptor variant by the Ensembl variant effect predictor, which is likely to have a significant impact on *DNAJB6* gene function (new Supplementary File 4). We added this data to the main text (Lines 320-328).

(2) We performed targeted genetic screening of the translated sequence of *DNAJB6* in DNA of 162 human SSS patients enrolled in Xinhua hospital, Shanghai Jiaotong University School of Medicine, Shanghai, China, for possible disease association. We identified one variant from one patient, as shown in the Response-only Table 1. Because our current evidence is still relatively weak – more pathogenic variants would need to be identified and functional validation of their pathogenicity need to be performed, we decided to not include these data in the present manuscript.

Taken together, in the Discussion section, we stated that “Prompted by our preliminary success in identifying potentially significant sequence variances for *DNAJB6* from human SSS patients, future human genetic studies are warranted to search more sequence variants and to confirm their pathogenicity, which are required to firmly establish *DNAJB6* as a new human *SSS* causative gene.” (Lines 391-395)

Reviewer #3 (Recommendations for the authors):1. The authors stated that Dnajb6b mutants were selected for in-depth characterization because they were most arrhythmogenic (lines 155-156). This is inconsistent with the data shown in Table 2.

In addition to increased incidence of SA, the *GBT411/GBT411* homozygous mutant also showed a significantly reduced heart rate, which does not occur in the other 2 candidate SSS mutants. Since the *GBT103* and *GBT411* mutants manifested similar tendency in terms of heart rate reduction, we removed the previous claim of *GBT411* as the most arrhythmogenic line and edited the abstract accordingly (Lines 45-46, 157-159).

2. The authors stated that Dnajb6b mutants' "response to isoproterenol remained unchanged" (lines 164-165). Based on what is presented in Supplemental Figure 1, Dnajb6b mutants responded to isoproterenol treatment in a manner like what was observed in wild type animals.

Thank you for pointing out this. We modified our statements by stating that “while its response to isoproterenol treatment appeared to be similar to that in WT control animals” (Lines 168-169).

3. The cardiac expression of Dnajb6 needs further clarification.a. The authors stated that Dnajb6 is enriched in SAN tissues (page 8). However, Dnajb6 appears to be expressed ubiquitously in the myocardium without obvious enrichment in SAN cells in fish (Figure 2).

In zebrafish myocardium, the expression of DNAJB6b indeed appears to be ubiquitous, unlike its more specific expression pattern in mouse SAN. Nevertheless, when we tried to define its expression in the conduction system through crossing into the sqET33-mi59B transgenic line in which EGFP labels the zebrafish SAN and atrio-ventricular canal (AVC) cells, we found the EGFP reporter partially overlapped with the Dnajb6b-mRFP fusion protein in the *GBT411* mutant during both embryonic and adult stages (new Figure 2)

b. Figure 2D, DNAJB6 is expressed in a broad range of cells which includes HCN4 positive cells. This is contradictory to what is stated in the abstract (lines 44-49) and discussion (lines 334-336).

Thank you for pointing out this error. Indeed, DNAJB6 antibody staining positive cells do partially, despite not completely, overlap with HCN4 positive Cells. We edited the abstract (Lines 48-49) and discussion (Lines 396-399) accordingly.

c. The authors highlighted cells with high TBX3/low DNAJB6 and low TBX3/high DNAJB6 expression (Figure 2E). Could the authors explain how high/low expression is defined, how many cells have been examined and what percentage of cells showing high TBX3/low DNAJB6 and low TBX3/high DNAJB6 expression?

We used the Zeiss Zen software equipped with the Zen2 microscope system to quantify the DNAJB6 and TBX3 signal intensity after antibody co-immunostaining. Our results (n=20 cells) showed a negative correlation between the DNAJB6 and TBX3 signal with an R^2^ of 0.8145 (new Figure 3E).

d. Figure 2F. islet1 staining shows high background.

Indeed, the Islet1 antibody immunostaining staining signal showed high background in our hands. We have tried 3 more commercially available antibodies (from abcam, catalog#: ab109517; from Developmental Studies hybridoma Bank (DSHB), catalog# 39.3F7 and 40.3A4) using SAN tissues dissected from adult mice hearts, but failed to obtain better images. To improve stringency of this manuscript, we took out of those Islet1 staining results and modified the text accordingly.

4. The impacts of DNAJB6 loss on heart rhythm in mice was examined in heterozygotes because Dnajb6-/- died embryonically. There are some issues regarding these mutants that need to be clarified:a. Could the authors clarify what stage do the homozygotes die and what stage are the hearts taken from mutants for Western blot? (Lines 213-216, Figure 3C).

The *Dnajb6* homozygous mice died in uterus at about E13.5 stage. We did Western blot using protein lysates extracted from the E12.5 stage embryonic hearts after PCR genotyping analysis. We clarified these details in the main text according (Lines 229-233, 563-569).

b. In mice, sinus arrest and AV block were noted in Dnajb6 +/-. Could the authors provide quantification? How many animals were examined? What percentage of animals develop sinus arrest and what percentage with AV block?

15 out of 44 the *Dnajb6+/-* mice examined manifested sinus arrest phenotype and 3 out of 44 exhibited av block, accounting for 33.4% and 6.8%, respectively. we now provide animal numbers and percentages of animals exhibiting sa or avb phenotypes in the *Dnajb6+/-* mutants in table 4.

c. Optical mapping revealed ectopic pacemaker activity in Dnajb6 +/- hearts. Could authors provide quantitative data?

We quantified the results shown in Figure 5B and added the *p* value. We used Fisher's exact test for statistical analysis of pacemaker redistribution in *Dnajb6* heterozygous mice and found *p*=0.0395 for the observed pattern of pacemaker distribution between WT and Dnajb6 mice. The Figure and corresponding Figure legend have been modified accordingly.

5. What is the cellular basis of ectopic pacemaker activity in Dnajb6 mutants? Are pacemaker cells damaged in Dnajb6 mutants?

Based on the diffused AV canal signal and SAN signal loss in the *GBT411/dnajb6* homozygous mutant, and a negatively correlated expression of DNAJB6 with TBX3 in the mouse SAN tissues, we speculate that DNAJB6 might acts as a suppressor of TBX3 transcription factor to define SAN cell specification. This potential mechanism is also supported by the observation in mice that loss-of-function of Dnaj6b results in conduction system defects and ectopic pacemaker activity. Since TBX3 suppresses chamber myocardial differentiation (Mommersteeg et al., Circ Res. 2007;100(3):354-62), upregulation of TBX3 may thus contribute to enhanced atrial ectopic activity in DNAJB6 heterozygous mice.

Furthermore, TBX3 has been recently identified as a component of the Wnt/β-catenin-dependent transcriptional complex (Zimmerli et al., *eLife*. 2020;9:e58123), which is significantly affected in Dnajb6 heterozygous mice (see new Figure 7B-C). This further supports a possible role of TBX3 in both SAN and atrial remodeling.

We did Trichrome Masson’s staining using SAN tissues from the Dnajb6 heterozygous mice and WT controls but didn’t detect any significant fibrosis changes (new Figure 5—figure supplement 2). Therefore, it is likely pacemakers are not damaged but their expression patterns altered in the Dnajb6 mutants.

All the observed ectopic pacemakers were located within the region of extensive distributed system of atrial pacemakers (the atrial pacemaker complex), which includes but extends well beyond an anatomically defined SAN (Boineau et al., Circulation 58: 1036–1048, 1978; Boineau et al., Circulation 77: 1221–1237, 1988; Glukhov et al., Am J Physiol Heart Circ Physiol 299:H482-H491, 2010). Cells located in the region between the superior vena cava and atrio-ventricular node and between the crista terminalis and inter-atrial septum, show a positive expression of pacemaker cell marker HCN4 (see Author response image 2). While the SAN cells are characterized by HCN4-negative and Connexin43 (Cx43, working myocardium-specific gap junction protein)-negative expression pattern, subsidiary atrial pacemakers have both HCN4- and Cx43-positive expression. In normal physiological conditions, spontaneous activity of subsidiary pacemakers is overdrive suppressed by the SAN. However, when SAN function is diminished (i.e., during SSS, as observed in Dnajb6 heterozygous mice), subsidiary pacemakers can produce escape beats leading to pacemaker irregularities and significant heart rate lability.

Though these subsidiary pacemaker clusters can provide a relatively regular rhythm, they are characterized by a slower resting heart, slower exertional heart rates, a prolonged post-pacing recovery time (a parameter similar to SAN recovery time but for non-SAN pacemakers), and an increased beat-to-beat heart rate variability (Morris et al., Cardiovasc Res 2013, 100, 160–169; Choudhury et al., J Physiol 2018, 596, 6141–6155). Furthermore, the electrical activity of this subsidiary pacemakers is more akin to that of the SAN than to the surrounding atrial muscle; the subsidiary pacemaker action potential exhibits prominent diastolic depolarization and a significantly lower maximum diastolic potential, take-off potential, overshoot, rate of rise, and amplitude than typical atrial muscle (Rozanski et al., JACC 1984, 4, 535–542). Finally, while being bradycardic in general, subsidiary atrial pacemakers can contribute to the development of atrial tachycardia (Kalman et al., JACC 1998, 31, 451–459; Marrouche et al., JACC 2002, 40, 1133–1139). Therefore, we could hypothesize that TBX3 overexpression observed in Dnajb6 heterozygous mice, could further facilitate pacemaker activity in cells within the extended atrial pacemaker complex and, maybe, promote atrial arrhythmogenesis in the setting of profound structural remodeling. These suggestions have been included in the revised Discussion section.

**Author response image 2. sa2fig2:** 3-D reconstruction of the entire mouse SAN preparations was used for immunofluorescent labeling of connexin 43 (Cx43, green) and HCN4 (red). Mosaic images were created by superimposing of 3-D Z-stacks (averaged through 30-50 scanned layers; 50-200 μm thickness) of different regions throughout the SAN pacemaker complex. Pacemaker regions are shown to be characterized by HCN4-positive staining. From Glukhov et al. Eur Heart J. 2015 Mar 14;36(11):686-97.

6. Transcriptomic profiling.a. Figure 6A. Missing log2FC value and p value for Gjb5.

We add log2FC value and *p* value data back for the *Gjb5* gene.

b. Heatmap shown in sup. Figure 3B is quite unusual. Is it scaled?

We double checked the heatmap and confirm the scale is correct. We used the function “Pheatmap” in Deseq package with default parameter settings, i.e., the values was centered and scaled within each sample. We do share the same concern that the order to show DE gene in “*Dnajb6^+/-^*” and “WT” is somewhat unusual. We tried to switch it but couldn’t, because the program automatically did it during the clustering process.